

**The role of siliceous sponges in pre-Eocene marine Si cycle from the perspective rock**
**mineralogy**
**Agata Jurkowska[1*], Ewa Świerczewska-Gładysz[2], Szymon Kowalik Filipowicz[3]**
[1] AGH University of Krakow, Poland, Faculty of Geology, Geophysics and Environmental Protection,
Mickiewicza St., 30, 30-059 Krakow, Poland; jurkowska.a@gmail.com; https://orcid.org/0000-0001-
5457-9968; Corresponding Author
[2] University of Łódź, Faculty of Geographical Sciences, Department of Geology and Geomorphology,
Narutowicza 88, 90-139 Łódź, Poland; ewa.swierczewska@geo.uni.lodz.pl; https://orcid.org/0000-
9 0003-4628-2712
[3] szymon.kowalik@10g.pl, independent researcher
**Abstract:** The process of siliceous sponge dissolution during diagenesis was interpreted not
only as an important part of marine Si cycle (comprising Si burial) but also as a significant
mechanism of chert formation (so-called "chertification"; Maliva and Siever, 1989a). Both
ideas were widely accepted by researches and are commonly used in geological studies. New
research contradicts these seminal assumptions and indicates that in pre-Eocene marine Si
cycle, although siliceous sponges were an important part of the ecosystems, did not play a
controlling role in regulating dSi (= dissolved silicon) concentration in the porewater as well as
in chert formation. The presented studies based on advanced mineralogical (XRD, EBSD;
SEM-EDS) and microtextural (SEM) analysis of rocks and sponge remnants verify the role of
siliceous sponges in the formation of Cretaceous siliceous rocks, by studying successions
deposited in similar marine environments, which contain abundant fossils of siliceous sponges
associated with cherts and authigenic silica polymorphs and those without them. For the first
time, the mineralogical and microtextural transformations of siliceous sponge loose
spicules/rigid skeletal networks, which led to their preservation as siliceous or pyrite/marcasite
infillings and also in form of limonite coatings, are presented. The data presented here about
the diagenesis of siliceous sponges skeletons opens the discussion on the usefulness of stable
isotopic studies of $\delta^{30}$Si in geological studies of fossils of silicifiers preserved as secondary
silica polymorphs (opal-CT).





**Keywords:** Si cycle; cherts; opal-CT; $\delta^{30}$Si, spicule diagenesis

## 1. Introduction

The diagenetic dissolution of skeletal elements of siliceous sponges in seabed mud is an
important part of the biogeochemical Si cycle in marine environments comprising the Si burial
in sediments via the rock formation (Maliva et al., 1989; Maliva and Siever, 1989a, b; Siever,
1991). According to the classical model of chert formation ("chertification"), this process is
directly controlled by the dissolution of siliceous sponge skeletons, which are the main source
of dSi (=dissolved silicon) for porewater (Maliva and Siever, 1989a; b). Moreover, the
importance of the process of siliceous sponge spicules dissolution is highlighted by the fact that
it has been interpreted as a basic mechanism controlling the porewater dSi concentration
(Maliva and Siever, 1989a, b; Siever, 1991). Recent studies have critically verified the previous
model of chertification and the biotic origin of siliceous nodules, and revealed that the dSi
seawater concentration governing the process of silica polymorphs precipitation and chert
formation mainly depends on environmental abiotic factors (volcano-hydrothermal dSi sources)
(Jurkowska and Świerczewska-Gładysz, 2020a, b; 2024 Jurkowska, 2022). This indicates that
previously accepted assumptions of a significant role of siliceous sponge dissolution in a Si
cycle need to be better recognized in order to evaluate its importance in controlling dSi
porewater concentration.
The key point of this research is the reconstruction of the process of early diagenetic dissolution
of biogenic opal-A of siliceous sponge spicules within the seabed mud followed by the silica
polymorphs transformation leading to the formation of siliceous sponge fossils and siliceous
(cherts) or carbonate-siliceous (opoka) rocks and environmental factors controlling them. The
geological record of siliceous sponge fossil and siliceous rock enables the reconstruction of



mineralogical silica phase transformation as well as factors controlling them, by mineralogical
and microtextural studies (Jurkowska and Świerczewska-Gładysz, 2020a, b; Jurkowska, 2022).
This analysis cannot be replicate in laboratory conditions due to their duration of tens of
thousands of years (Kastner et al., 1977). The studies have been conducted in Upper Cretaceous
deposits of the European Basin and present different models of siliceous sponge dissolution and
silica polymorphs reprecipitation reflected in various types of preservation sponge skeletons in
a carbonate/clayey seabed mud.
The studies presented here have an impact not only on the recognition of rocks genesis but also
for evaluation of the usefulness of the rapidly developing scientific tool of $\delta^{30}Si$ measurements
of siliceous sponge spicules built of primary opal-A for the estimation of seawater dSi
concentrations (Sutton et al., 2018; Hendry, 2012). This very promising idea, which is used for
to determine dSi in recent and Cenozoic oceans, could potentially be transferred into Paleozoic-
Mesozoic fossil siliceous sponges. However, without proper recognition and documentation of
a diagenetic process of sponge spicules dissolution and biogenic opal-A transformations its
usefulness in paleontological record is doubtful. Taking into account that the process of
siliceous sponge skeleton dissolution is a complex series of environmentally dependent
polymorphic silica transformations, during which the primary signature of $\delta^{30}Si$ can be
changed, the detailed recognition of the mechanism of skeleton dissolution followed by silica
reprecipitation is essential for the use for of $\delta^{30}Si$ from skeletons of fossil sponges in the
geological interpretations.

**2.       Seminal model (Maliva and Siever, 1989a): the role of siliceous sponge dissolution**
**in siliceous nodules and carbonate-siliceous rock formation**
The general assumptions about the role of siliceous sponges in the formation of siliceous bedded
and nodular cherts and flints have been summarized and presented by Maliva and Siever



(1989a, b), who described the model of siliceous nodule formation in Mesozoic and Cainozoic
deposits (Fig. 1). All the assumptions that laid the foundations for Maliva and Siever (1989a,
b) model of 'chertification' are derived by mineralogical and paleontological analysis of
geological record of field sections (Bromley and Ekdale, 1984; Clayton, 1984) and DSDP
(Deep Sea Drilling Project) cores (Heath and Moberly 1971; Von Rad and Rosch, 1974; Wise
and Weaver, 1974). The classical model of the formation of siliceous cherts and flints, which
has been generally accepted in the literature and extended for other carbonate siliceous rocks
(e.g. opoka, gaize, Sujkowski, 1931; Pożaryska, 1952), describes this authigenic process as a
complex series of dissolution/precipitation reactions trigged by changes of geochemistry within
the seabed mud and followed by later diagenetic silica polymorphs maturation, which finally
led to formation of siliceous rocks. The presented model involves four main stages of silica
polymorphs precipitation (Fig. 1) in combination with the assumption that essential for the silica
polymorphs precipitation elevated dSi seawater concentration (at least of a level 10–60 ppm;
250–1000 µM; (Siever, 1991)) was achieved due to siliceous sponge (opal-A) dissolution.
The general argument presented in the literature confirming that the siliceous sponge spicules
have been the main source of dSi for siliceous rock formation is based on the observation of a
presence fossils of sponge spicules and/or voids left after their dissolution within the siliceous
nodules/carbonate siliceous rocks. In many studies, although the sponge fossils are very rare,
they were treated as an argument confirming the assumption of biogenic dSi origin. The
assumption of biogenic dSi origin was so grounded even with the absence of siliceous sponge
remains in the rocks, it has been explained by its complete dissolution (Maliva and Siever,
1989a, b). The other arguments that have been used to confirm biogenic origin of dSi for
siliceous rock formations was based on a correlation between facies chert distribution and
siliceous fauna migration in Earth history (Maliva et al., 1989). This idea has been critically
discussed in our previous article (Jurkowska and Świerczewska-Gładysz, 2024) and in light of





new data summarized there, this argument can no longer be used to confim the biogenic origin
of siliceous nodules. The additional statement assumed in seminal work of Maliva and Siever
(1989a, b) concludes that besides the siliceous skeletons, there is no probable source of dSi in
seawater. This last argument was also critically discussed (Jurkowska and Świerczewska-
Gładysz, 2024 and literature cited therein) and, based on new data, the dSi source of
volcanic/hydrothermal origin connected with LIP (Large Igneous Provinces) activity was
pointed out. To sum up from the main three arguments which has been used to confirm the
biogenic origin of cherts in the classical model (Maliva and Siever, 1989a, b) the mechanism
of siliceous sponge spicules dissolution has not been tested by using new analytical techniques
as a potentially controlling factor of dSi concentration in porewater.

**3.       Siliceous sponge – sediment system an methodological approach**
The role of siliceous sponges in the chertification process involves the siliceous skeleton
dissolution making them a source of dSi to the seawater, which directly controls the overall dSi
concentration and is followed by abiotic silica polymorphs precipitation (Jurkowska, 2022).
The process of biogenic opal-A dissolution and silica precipitation is geochemically dependent
and involves chemical interaction between the external environment and distinctive
microenvironment that forms around and inside of the decaying sponge. The geochemical
environmental factors which control the silica dissolution and its reprecipitation could be
characterized as dSi concentration of seawater/porewater, organic matter (OM)
content/distribution and mineralogical composition of the seabed mud (content of carbonates
and clays) (Williams and Crerar, 1985; Kastner et al., 1977). Sponge morphology (presence of
loose or fused spicules, wall thickness) and organic matter content control the ability to the
chemical interaction between the external environment and microenvironment of decaying
sponge remnants, involving the possibility of the dSi migration from the siliceous skeleton



dissolution to the seawater/porewater (Jurkowska and Świerczewska-Gładysz, 2020a; b;
Jurkowska, 2022). Assuming the process of siliceous sponges skeleton dissolution is dependent
on geochemical conditions of the surrounding environment and interaction of a porewater with
decaying sponge body to recognize the complex mechanism of the sponge siliceous skeleton
dissolution and silica reprecipitation, and environmental and internal (sponge morphology)
factors controlling these processes, we designed the mineralogical/microtextural studies of
various sponge fossils.
The methodological background for testing the role of siliceous sponge dissolution in the
formation of siliceous rocks is based on a selection of siliceous rocks (cherts) and carbonate-
siliceous (opoka) sedimentary rocks deposited during a time interval in which the siliceous
sponges have been indicated as a source of dSi for their formation (Calvert, 1974, 1977; Maliva
and Siever, 1989a; b; Siever, 1991). For comparative analysis, we choose rocks (limestones and
marls) in which the fossils of siliceous sponges are common, but the silica polymorphs have
not been detected in previous geological studies. We chose the successions deposited under
similar marine conditions (carbonate environment) during a relatively short stratigraphic range
(Upper Cretaceous: upper Turonian-lower Coniacian, lower-middle Campanian and upper
Campanian-lower Maastrichtian) and in the small geographic area of one depositional
epicontinental basin (southern part of the Polish Basin).
The primary environmental factor that affects the potential for siliceous sponge skeleton
dissolution and silica precipitation is seawater dSi concentration (Kastner et al., 1977; Williams
and Crerar, 1985). Generally, during the Earth history, seawater dSi concentration was below
the threshold opal-A precipitation (<1200 μM) (Siever, 1991; see also Jurkowska and
Świerczewska-Gładysz, 2024) which means that seawater chemistry promoted siliceous sponge
dissolution, indicating that siliceous skeletons potentially could be a good source of dSi in
seawater/porewater for siliceous rock formation. To test the impact of different dSi





concentration on the rate of siliceous sponges dissolution and its reprecipitation, we chose the
Campanian-Maastrichtian sections of opoka intercalated with cherts and marls, for which
different dSi concentration during deposition have been indicated (Jurkowska and
Świerczewska-Gładysz, 2020a; b). We extended the analysis to the upper Campanian-lower
Maastrichtian sections in which the opoka was deposited in a similar environment and have
asimilar mineralogical composition to the aforementioned opoka samples (see opoka
mineralogical classification in Jurkowska, 2022; Jurkowska and Świerczewska-Gładysz, 2022),
but is not interlayered with cherts and contains significantly more abundant voids left after
spicule dissolution, dispersed in a rock matrix and body-preserved sponges, whose skeletons
are also usually dissolved and pieces of siliceous network are rarely preserved (Jurkowska and
Świerczewska-Gładysz, 2020b).
Another factor that affects the siliceous sponge decomposition and dissolution of its skeleton
in sediment, as well as the silica polymorphs precipitation during diagenesis, is the presence
and distribution of organic matter (OM) within the seabed mud. The process of OM
aerobic/anaerobic microbial decomposition in the sediment column (so-called redox-cascade)
controls the geochemistry of the environment and triggers the diagenetic changes through
fluctuations in the pH and Eh conditions of porewater circulating within the seabed mud. This
is crucial for silica polymorph precipitation, for which the available alkalinity and $Mg^{2+}$ ions
are essential and can only be achieved through microbial activity within the seabed mud
(Zjistra, 1987, 1994; Jurkowska, 2022; Meister et al., 2022). Under extremely oligotrophic
conditions, where the amount of OM in the seabed mud is decreased, the rate of microbial
decomposition of OM does not trigger the geochemical changes essential for alkalinity
production necessary for silica polymorphs precipitation. If the decomposition of OM starts
within interspicular spaces (which are cut off from the external environment), it can create a
microenvironment inside these spaces that is geochemically different (e.g. anaerobic



conditions) from surrounding seabed mud (Jurkowska and Świerczewska-Gładysz, 2020a). In
this study, the recognition of the influence of OM presence and distribution on dissolution of
siliceous sponges will be tested by studying specimens from the upper Turonian limestone for
which oligotrophic conditions were indicated (Jurkowska et al., 2018; Płachno et al., 2018), and
upper Turonian-lower Coniacian marls, in which the close microenvironment of decaying
sponge remnant will be analysed.
The last significant environmental factor which strongly affects siliceous sponge dissolution
and controls the silica polymorph precipitation is the primary (non-authigenic) mineralogical
composition of the sea bottom mud (Isson and Planavsky, 2018). In the studied sections, which
represent a relatively monotonous mineralogical composition (Jurkowska, 2022), this
parameter is expressed in the variable content of detrital clays. Clays affect significantly the
rate of silica polymorph dissolution and nucleation during diagenesis by retarding of the whole
process and by scavenging dSi and free space available for precipitation from newly formed
silica phases (Kastner et al., 1977; Isson and Planavsky, 2018; Jurkowska, 2022).
Non-environmental factors which also have a significant impact on the dissolution of silica
spicules and their preservation are features of sponges such as the structure of the skeleton of
individual sponges and the amount of soft body, which influences the quantity of organic matter
delivered by the dead sponge to the sediment. The rigid skeletal network of some sponges
favoured its preservation as bodily preserved fossils and created the a close microenvironment
around the decaying organic remnants. This rigid skeleton consists of articulated spicules called
desmas, which are found in some demosponges (belonging to the informal lithistida group), or
is formed by fused spicules (hexactins or lychniscs), occurring in two groups of hexactinellids
(Hexactinosida and Lychnicosida). In other groups of hexactinellids and demosponges, the
skeleton consists of loose spicules that disperse after the breakdown of the soft body. The
amount of soft body is mainly related to the thickness of the sponge wall. Most of the




demosponges noted in the studied section are represented by conical, cylindrical or bulbous
bodies with thick wall. In contrast,, hexactinellids are dominated by vase-, plate- or tube-like
specimens with very thin walls (their thickness is usually 2–3 mm, and only occasionally
reaches 5 mm).

**4.    Materials and methods**
All studied sections were deposited in Late Cretaceous European Basin under the carbonate
pelagic conditions of low terrygenious influx (opoka, cherts) and under the input of detrital
clays (limestone, marls). The oldest studied rocks are upper Turonian limestones (of the Marly
Limestone Unit) and upper Turonian-lower Coniacian marls (of the Upper Marls Unit) forming
ca.. 35 m section of Opole area (e.g. Alexandrowicz and Radwan, 1973; Walaszczyk, 1988;
Kędzierski and Uchman, 2015; Świerczewska-Gładysz et al., 2019) (Tab. 1). These deposits
are exposed in the Folwark quarry, where paleontological and taphonomic studies of siliceous
sponge remains were conducted in two lithologies: marls and limestones. To confirm the
macroscopic observation of lithological type the quantitative and qualitative XRD analyses of
rocks mineralogical composition has been performed. The results indicate that the limestones
are mostly composed of calcium carbonate (94%), clays (illite-smectite) (5%), and an
insignificant amount of detrital quartz (1%). The marls can be distinguished from limestone by
lower calcium carbonate content (84%), higher clays (illite-smectite) (11%), and detrital quartz
content (5%). The studied upper Turonian-lower Coniacian carbonate rocks do not contain
silica polymorphs as a rock component, although siliceous sponge remnants are common.
Sponge assemblages from this section are dominated by body preserved hexactinellid sponges
with rigid skeleton, the body preserved lithistids are extremely rare, while rare siliceous sponges
without rigid skeleton are noted mainly in marls as a loose spicules or their moulds



(Świerczewska-Gładysz, 2012a; Świerczewska-Gładysz and Jurkowska 2013; Świerczewska-
Gładysz et al., 2019).
The other types of rocks that have been analysed are lower-middle Campanian (the rock which
contain opal-CT forming a siliceous framework structure (Jurkowska and Świerczewska-
Gładysz, 2022) of Miechów Synclinorium (MS) and middle Campanian-lower Maastrichtian
opoka middle Vistula River valley (MVR), lower-middle Campanian cherts (the siliceous
nodules primary composed of opal-CT) and marls of MS. The mineralogical composition of
these rocks of both regions has been studied in a previous works (Jurkowska and Świerczewska-
Gładysz, 2020a, b) and summarized in Jurkowska (2022; Tab. 1). Observation and sampling
was carried out in several outcrops: Dziurków, Piotrawin, Raj N., Pawłowice Cm, Dorotka
(MVR), Wierzbica, Pniaki, Jeżówka 2, Rzeżuśnia, Biała Wielka (MS) (Tab. 1). All studied
rocks contain a variable amount of authigenic opal-CT, which indicates the early diagenetic
precipitation of silica polymorphs and remnants of siliceous sponges. Spicules of non-lithistid
demosponges or voids after them are very numerous in opoka (Świerczewska-Gładysz 2012b;
Jurkowska and Świerczewska-Gładysz, 2020a, b), but are also present in marls and cherts
(Jurkowska and Świerczewska-Gładysz, 2020b). The macrofossils of sponges in studied
sections are represented by body-preserved specimens of hexactinellids and demosponges with
rigid skeletal network (e.g. Bieda, 1933; Hurcewicz, 1966, 1969; Świerczewska-Gładysz, 2006,
2016), and rare remains after large loose spicules of hexactinellids without rigid skeletal
network (Świerczewska-Gładysz and Jurkowska, 2013). The fossils have been collected from
the entire succession in each section, while microscopic analysis of sponge remains has been
performed on samples collected from each section at intervals of 1m from lithologically
monotonous sections and from different rock types if the lithological difference has been
observed (Tab. 1).



The second line of studies comprised microtextural studies, which were based on detailed SEM
observations (using an FEI QUANTA 200 scanning electron microscope) in three modes: SE
(secondary electrons), backscattered electrons (BSE) and a mix (combination of both previous
modes) to analyse the state of preservation and microtexture of siliceous sponges remnants
(loose spicules or spicules forming rigid skeletal network). The SEM observations have also
been performed on rock surrounding the sponge remnants and rock chips representing the
lithology from which the fossils originate. From the last ones also the insoluble residuum of 0.5
cm and 0.63 fraction has been analysed. The insoluble residuum has been prepared according
to the protocol describe in Jurkowska et al. (2019).
The second line of studies comprised the mineralogical recognition of sponge remnants, which
was realized through SEM-EDS, XRD, and EBSD analyses. For those samples from which
there was a possibility to gain enough material to perform XRD studies, those analyses have
been performed. The SEM-EDS analysis combined with XRD data has been used to recognize
the distribution and microtextural features of minerals representing polymorph groups. The
EBSD studies have been performed to trace the distribution of iron sulfides (pyrite and
marcasite) and barite within the sample. For these analyses, the polished sample surface was
examined in a scanning electron microscope FEI Versa 3D (FEI Company, Hillsboro, Oregon,
USA). The SEM observations were conducted using an electron backscatter diffraction (EBSD,
Symmetry S-2, Oxford Instruments Nanoanlysis, High Wycombe, UK) detector. To prevent
sample charging, the microscope was operated at 20 kV and ~12 nA in low vacuum mode at 20
Pa. The EBSD detector was operated in "Speed 2" mode, with a 156x128 pixel resolution and
around 300 patterns per second. Aztec (ver. 6.1, Oxford Instruments Nanoanalysis) software
was used to index the diffraction patterns. Hough transform resolution was set to 70, and 12
detected bands were used for indexing. The presented maps are "raw data" and were not
subjected to any cleaning or modification.



**5.     Results and comments**
*5.1 The state of preservation and mineralogy of sponge skeletons*
In all studied lithologies, the loose spicules and skeletal network that are preserved as siliceous
remains are composed in different ratios of opal-CT (6–82%) and nano-α-quartz. The quartz
that builds the sponges' skeletal components show a smooth and uniform microtexture (Fig.
2a). On the surface of siliceous spicules (loose or forming a rigid skeletal network) different
types of dissolution remarks are visible, as cavernous pattern, rounded and platy dissolution
remarks (Fig. 2b). Usually, the external nano-α-quartz surface is overgrown by a single, dense
layer of opal-CT with well-visible blades (Fig. 2c) and/or early forms of embryonic opal-CT/"
Mg-rich clays" (Jurkowska and Świerczewska-Gładysz, 2020a) or a mixed opal-CT clayey
layer (Fig. 2d). The rounded dissolution features are secondaryily infilled by single lepispheres
of opal-CT of variable sizes (10–60 μm) and shapes (rounded to mushroom-like shapes) (Fig.
2e). Those opal-CT lepispheres show smooth microtexture without visible crystallized blades
and probably represent early forms of opal-CT, while their shape is controlled by the free space
available for precipitation. The space inside the spicules could be infilled by smooth lepispheres
of opal-CT (Fig. 2a) and porous or smooth silica, probably of quartz composition (Fig. 2f), or
homogeneous dense mass of mixed opal-CT and nano-α-quartz (Fig. 2g). Rarely, the only
preserved remnants are the spicule infillings composed of smooth opal-CT lepispheres
cemented by porous silica, while the external layer of nano-α-quartz is not preserved (Fig. 2h).
Another mineral that infills the voids left after the dissolution of siliceous sponge skeletal
elements is pyrite with subordinate marcasite. The microtexture of pyrite and marcasite
infillings is smooth on the outer part, while on the inside, the pyrite crystals are visible and the
void left after the dissolution of the area around the central canal is still preserved (Fig. 3a, b).
In some specimens, the external surface of the skeleton (consisting of loose ectosomal spicules
and/or the external part of rigid choanosomal network) represents two different mineralogies of



pyrite with marcasite with preserved remnants of siliceous (mixed opal-CT/nano-α-quartz)
spicules (Fig. 3c). In a few specimens, the voids left after the dissolution of the area around the
central canal are infilled by barite (Fig. 3b).
Euhedral pyrite crystals outline the voids left after the dissolution of loose spicules and the
skeletal network of siliceous sponges (Fig. 3d). The pyrite crystals are of cubic, pyritohedral to
octahedral crystal morphology, usually of uniform sizes (1–10μm). Rarely, pyrite framboids
are visible, of uniform sizes (3–6 μm), usually associated with very small (<1 μm) pyrite
crystals. In most samples, pyrite crystals show oxidation remarks (Fig. 3e) and corrosion pits
(compare: Chen et al., 2022) and are associated with visible lumps of organic matter (Fig. 3e).
Another type of mineralogy observed in studied paleontological material is ferrigenous
(representing various type of minerals of the limonite group: lepidocrocite, geothite, and
hematite) and mixed pyrite and ferrigenous coatings outlining the previously siliceous skeleton,
which is mostly dissolved (Fig. 3f). Preserved fragments of a siliceous skeleton built of mixed
opal-CT/nano-α-quartz structure are rare. A variety of different microtexture of ferrigenous
coatings is observed among this state of preservation and could be described as homogenous
mass forming a smooth texture inside and with a cavernous pattern on the outside of the sponge
spicule (Fig 3g), fibrous and fuzzy microtexture, and blocky microtexture (Fig. 3h). The pyrite
with limonite coatings is represented by euhedral pyrite crystals (sometimes forming
framboids) of uniform sizes, which are covered by layers of limonite minerals.
The remnants of the presence of siliceous sponges are also recorded as voids left after spicules
dissolution, preserved due to the existence of an external single-layer coating of silica, which
microtexturally resembles opal-CT. The voids left after spicules have the original shapes and
sizes of the spicules (Fig. 3i, j). Whether this layer is an external zone formed due to infilling
by secondary silica polymorphs of the original spicule or formed as a sediment layer
surrounding the original spicules is not known. This type of preservation is very common and



mostly represents megascleres of non-lithistid demosponges (Fig. 3i), but is also observed in
remains of other siliceous sponges (Fig. 3j) (Jurkowska and Świerczewska-Gładysz, 2020a, b).

*5.2 First factor: impact of dSi concentration on the rate of dissolution of sponge skeletons and*
*precipitation of silica polymorphs*
In Campanian cherts and chert nodules with flint cores (for the macroscopic description and
mineralogy details, see: Jurkowska and Świerczewska-Gładysz, 2020b) among the spicules,
oxeas and triaenes of non-lithistid demosponges dominate (Fig. 3k) while the loose hexactines
and root tufts of lyssacinosids, fragments of the rigid skeletal networks of lithistids (Fig. 3l),
and hexactinellid sponges (lychniscosidan and hexactinosidan) are less common (Jurkowska
and Świerczewska-Gładysz, 2020b). Most of the loose spicules and the fragments of the skeletal
network are siliceous (Fig. 3k, l) and are built of a single outer layer of homogeneous mass of
nano-α-quartz (sometimes covered by single fibrous layer of opal-CT (Fig. 2c) and internal
lepispheric opal-CT infillings (Fig. 2e, f), which are mineralogically and microtexturally
different from the surrounding cherts built of large lepispheres (20–30μm) of opal-CT or
siliceous rock network (Jurkowska and Świerczewska-Gładysz, 2020). The combination of
observations that the boundaries of chert nodules do not overlap the outlines of sponge
macrofossils and that silica polymorphs of chert nodule present different microtexture than
silica infilling the voids left after spicules, indicate that the process of chert nodule formation
was generally independent from the siliceous sponge skeleton dissolution and/or affected by it
to a small extend (Jurkowska and Świerczewska-Gładysz, 2020). Although the chemical
interaction and dSi migration between the dissolving sponge skeleton and surrounding mud
environment must exit, the dSi diffusion between the seawater and porewater was the main
process which controlled the dynamic balance of dSi concentration, which governs the opal-A
dissolution, silica precipitation inside the void left after skeleton dissolution and and the



formation of cherts. The infilling of the siliceous remnants of the sponge skeleton by secondary
authigenic silica polymorphs indicates that dSi migrates back into the voids left after skeleton
dissolution, probably after the chert precipitation or the chemical balance between the
dissolving siliceous skeleton and surrounding environment of chert was established at a dSi
level enabling for the authigenic silica precipitation inside the voids. Moreover, the existence
of dissolution remarks on a siliceous infillings of siliceous sponges indicates that after the
precipitation of siliceous infillings of voids left after spicules (Fig. 2b), the dSi level in
surrounding environment declined, triggering the further authigenic silica dissolution. The
dynamic interaction governing silica dissolution and precipitation was realized in the studied
system via dSi diffusion between three sites: seawater, cherts (by porewater) and opal-A sponge
skeleton, in which the first and the last acted as dSi source, while cherts and voids left after
spicule dissolution were dSi sinks. The process behind the dSi migration from the source to the
site of precipitation is Landmesser diffusion (Jurkowska and Świerczewska-Gładysz, 2020b,
2024) and indicates that although the siliceous sponge skeletons, through their dissolution, acted
as a source of dSi for the formation of siliceous nodules, seawater was the main source of dSi,
providing the constant dSi concentration at the level of opal-CT during the chert formation, as
well as precipitation of siliceous infillings of voids left after spicules (Jurkowska and
Świerczewska-Gładysz, 2020b). If the sponge skeletons were the main dSi source, such an
inflow would not be possible, and overall dSi concentration would be lower due to dSi diffusion
through the seawater, moreover, the siliceous infillings of voids left after spicules would not be
formed because after the spicule dissolution the whole process would stop.
The SEM observations conducted in this research also enable the reconstruction of the process
of the spicule dissolution followed by silica polymorphs precipitation inside the voids left after
spicules dissolution (Fig. 4). The process of dissolution started when the opal-A began to absorb
foreign ions and transformed into an intermediate form of opal-A' without any microtextural



changes. The dissolution started from a central canal within spicules and then their surface (see

e.g. Rützler and Macintyre, 1978; Bertollino et al. 2013, 2017; Costa et al. 2021), which led to

the complete dissolution of opal-A'. The voids left after spicule dissolution were imprinted in

rock by precipitating within the sediment opal-CT lepispheres (Fig. 4). In the studied

environment, all the essential factors necessary for silica polymorphs precipitation were

available. The dSi concentration, which is the primary factor necessary for silica polymorphs

precipitation, was available due to high seawater dSi concentration, while the other $Mg^{2+}$ions

and alkalinity were also present due to calcite and aragonite dissolution triggered by the pH

drop during the microbial decomposition of OM (Jurkowska and Świerczewska-Gładysz,

2020a; 2022). Inside the voids, the first phase to precipitate was an opal-CT, forming rounded

lepispheres of variable sizes and each of them acted as site and source for the diffusing dSi.

Between the lepispheres, when the dSi concentration dropped to the level of quartz precipitation

(10 ppm= 250 µM), the quartz started to precipitate, but its growth was prevented due to

simultaneously occurring dissolution initiated by dSi diffusion controlled by precipitating opal-

CT lepispheres (Fig. 4). When the dSi declined, the external layer of poorly formed quartz

crystallized, followed by the precipitation of a thin layer of early fibrous opal-CT forms or early

embryonic forms of opal-CT/"Mg-rich clays" under elevated dSi concentration or dissolution

(visible dissolution remarks) when dSi concentration declined.

*5.3 Second factor: the impact of organic matter content (OM) on dissolution of skeleton*

*remnants of siliceous sponges and silica polymorphs precipitation*

*5.3.1 The OM content and distribution in the sediment*

In the studied succession of Folwark quarry, the limestone of the Marly Limestone Unit was

deposited under oligotrophic conditions dominated by opportunistic organisms (calcispheres-

pithonellid assemblages – D. Rehakova pers. comm (Fig. 5a) (Jurkowska et al., 2018; Dias-



Brito, 2000) or those using the distinctive feeding strategy (*Lepidenteron mantelli* – Jurkowska
et al., 2018) to survive in an environment impoverished with biogenic elements. The
oligotrophic conditions are also well tolerated by hexactinellid sponges, which are the most
numerous group of fossils in these layers (Świerczewska-Gładysz, 2012a; Świerczewska-
Gładysz et al., 2019). Our observations indicate that in such an environment, diagenetic changes
(dissolution followed by authigenesis of newly formed minerals) were limited because the
overall decreased OM content did not trigger  effective fluctuations of the environmental
geochemistry during the microbial OM decomposition. In the rock matrix, the only observed
authigenic mineral phases are rare newly-formed small (10–15 µm) calcite grains of rounded
to subhedral shapes, which are incorporated into the coccoliths carbonate rock matrix (Fig. 5b).
The allomicritic grains of calcite did not show any signs of dissolution, indicating that only a
slight drop in pH to a level where the aragonite elements and small calcite grains undergo
dissolution took place during early diagenesis. The absence of silica polymorphs in the rock
matrix or infillings of voids after spicules dissolution could be caused by diminished rate of
OM in the sediment due to oligotrophic conditions, which have a limiting effect on silica
polymorphs precipitation. The sponge skeleton are preserved as mixed euhedral pyrite with
subordinate ferrigenous (limonite group) coatings outlining the previous siliceous skeleton (Fig.
5c, d, e). This indicate that although the siliceous sponge skeleton undergoes dissolution and
deliver the dSi to seawater/porewater and the precipitation of siliceous polymorphs did not took
place. The similar state of siliceous sponge fossil preservation, but containing only ferrigenous
coatings of limonite group (Fig. 5f), is also documented in opoka sections, however, in the last
the silica polymorphs (as opal-CT) are present in the rock matrix, as well as significant amount
of newly formed calcite (Jurkowska, 2022). This suggests that in the upper Turonian limestone,
although the siliceous skeletons undergo dissolution, cherts and opoka did not formdue to the
existence of a factor, that limits silica polymorphs precipitation. Taking into account that during





the Cretaceous the seawater dSi concentration was high (Siever, 1991), the diminished rate of
calcium carbonate dissolution (due to small amount of OM) and the related lack of available
alkalinity and $Mg^{2+}$ ions could be the cause of the lack of a new silica polymorph precipitates
(Williams and Crerar, 1985; Kastner et al., 1977). The latter is a very probable factor because
the time intervals of Turonian calcisphere blooms are correlated with events of unusually high
$Ca^{2+}$ concentration and low Mg/Ca ratio (Stanley et al., 2005; Van Dijk et al., 2016; Ciurej et
al., 2023) which could limit the availability of $Mg^{2+}$ ions essential for silica polymorphs
precipitation (Williams and Crerar, 1985; Kastner et al., 1977). Another factor that needs to be
taken into account is lower dSi seawater concentration, which could occur in some areas during
the Cretaceous (Jurkowska and Świerczewska-Gładysz, 2020a), but this aspect has never been
studied in Turonian limestone. The very abundant hexactinellid sponges documented in
Turonian limestone are similar to those noted in Campanian opoka deposited under relatively
high dSi seawater concentration (Jurkowska and Świerczewska-Gładysz, 2022) which
contradicts the diminished seawater dSi concentration during time of Turonian limestone
deposition and implies that similar conditions could have existed during the limestone
formation. The presence of pyrite and ferrigenous coatings only in a microenvironment
outlining the previous siliceous sponge skeleton and their absence in the surrounding mud is
also related to the presence of OM and its distribution within the sponge body. Although the
general content of OM in studied rocks is small, observation indicate that the OM is preserved
as lumps of homogeneous mass or fibrous texture associated only with sponge skeletons (Fig.
5f) and always occurs with pyrite crystals/limonite coatings. This association, which is quite
common in the paleontological record, is usually connected with distinctive reductive
conditions and acidic pH created by bacterial decomposition of OM in a small
microenvironment surrounding the organism remnants. Those characteristic conditions are
different from the geochemical conditions of the environment in the surrounding mud.



Comparing the preservation of siliceous sponge skeleton as ferrigenous coatings from opoka
(Fig. 3f-i) and limestone, the difference is attributed to the variable mineral phases that build
those coatings. In opoka, the siliceous sponge skeleton outline is formed of the limonite group
(Fig. 3f-i), while in limestone, pyrite crystals partially covered by limonite mass occur (Fig. 5c,
d), indicating that in the former, the oxic conditions prevailed during iron minerals formation,
while in the second the anoxic conditions prevailed. The oxidation pitches and dissolution
features visible on pyrite euhedral crystals (Fig. 5e), indicate a later oxidation event, which
could also cause the transformation of pyrite into limonite group minerals. This could be
connected with recent conditions, probably occurring during rock excavation or during late
diagenesis due to contact with oxygenated porewater.
In limestone, the presence of OM associated only with siliceous sponges (Fig. 5f) and with
pyrite mineralogy indicates that due to oligotrophic conditions, the OM underwent anaerobic
microbial decomposition, while not decaying in an oxic zone. This is caused by the relatively
small number of sponge-feeding predators (e.g. specialized snails, starfish) (e.g. Stratmann et.
al., 2022; López-Acosta et al., 2023). Additionally, under oligotrophic conditions, the rare
bottom-feeding organisms did not facilitate the sponge fragmentation. In the studied
environment, because of oligotrophic conditions, this process took place deeper within the
sediment column in the dysoxic/anoxic iron and sulphate reduction zone, where the pyrite was
formed due to availability of the essential substrates ($Fe^{2+}$ and $H_2S$) for its precipitation.
In the opoka environment, in which the oxic and eutrophic conditions prevailed, the OM on a
sponge body decomposed in an oxic zone, causing that iron oxides and hydroxides the
formation (Fig. 3f-h).
The formation of pyrite crystals, as well as limonite group minerals, needs iron availability in
the immediate vicinity of the sponge skeleton in an amount higher than its content in seawater.
The presence of a significant amount of iron ($Fe^{2+}$) essential for the formation of iron sulphides



478 (pyrite) and iron oxides/hydroxides around the sponge remnants could be associated with the

479 organism's biomineralization process during which the iron was actively accumulated in a

480 living sponge body (Gentric et al., 2016; Kubiak et al., 2023). The silanol groups (Si-OH) of

481 siliceous sponges can also bond $Fe^{2+}$ to the structure by replacement of OH groups. The $Fe^{2+}$

482 (Ferretti et al., 2019) is released during siliceous skeleton dissolution and saturates the

483 microenvironment. Moreover, observations of modern sponges have shown that the skeletons

484 of dead sponges, as a result of long-term contact with seawater/porewater, become covered with

485 a coating containing metal ions, including iron ions, which are incorporated in into structure of

486 amorphous silica (Hurd 1973; Chu et al., 2011). These iron ions come from seawater/porewater

487 and are attracted by the negatively charged surface occurring on the silica sponge framework.

488

489 *5.3.2 The OM content in a closed microenvironment of the decaying sponge body*

490 The state of siliceous sponge preservation in the upper Turonian-lower Coniacian marls is

491 significantly different from that observed in the limestone (Chapter 5.3.1), which is caused by

492 the formation of a geochemically different, closed microenvironment inside the decaying

493 sponge compared to the surrounding sediment. Trapped OM within the sponge came from

494 sponge's body, microbes decomposing it, and also from their symbiotic microbes (Kluijve et

495 al. 2021) and microbes that decompose the sponge's body (Stratmann et al. 2022). Within the

496 state of preservation of sponge skeletons from these marls, the two types related to sponge

497 groups, their morphology, and distribution of OM within the spaces in the skeleton could be

498 distinguished: (1) the rigid (choanosomal) skeleton of hexactinellid and lithistid sponges

499 preserved as pyrite infillings with marcasite (Fig. 3a, b; 6a) and barite (only in lithistids) (Fig.

500 6b) and the surface part of the skeleton (loose ectosomal spicules and/or external choanosomal

501 network) preserved as siliceous (Fig. 3c, 6b); (2) the loose spicules of body-preserved non-rigid

502 demosponges with dissolved siliceous skeleton occurring as pyrite and marcasite infillings



associated with barite (Fig. 3b, 6c, e, g, h). The typical feature of both these types of
preservation is the presence of carbonate sediment occurring inside the spaces between spicules
(loose, articulated, or fused) in the form of authigenic small grains of homogeneous
coalescence/fused texture (Fig. 6d) and large sparite crystals forming cements (Fig. 6e). These
texture were formed due to complete primary calcite dissolution followed by its reprecipitation
in the sediment infilling the sponges. Such authigenic grains were not documented in the
surrounding carbonate mud (Fig. 6f), indicating that the closed microenvironment must have
developed inside the decomposing sponge body and governed geochemical conditions different
from those in the surrounding environment. The formation of a closed microenvironment cut
off from surrounding porewater probably started before sponge burial and was generated by
microbial biofilm formed around the decaying sponge remnants (compare Stratmann et al.,
2022) (Fig. 7). After burial, the chemical barrier between the decaying sponge and surrounding
sediment was also facilitated by the high clay content within the seabed mud, which formed a
physical barrier (Fig. 6f).
In both types of preservation, the anoxic and acidic pH conditions generated inside the sponge
body due to OM decomposition and cutting off from external oxic environment trigger the
dissolution of opal-A in sponge spicules and crystallization of small euhedral pyrite crystals in
space between spicules, as well as the precipitation of massive polycrystalline pyrite infillings
in voids left after spicules (Fig. 3a, b, 6a, g-i) (compare Reolid, 2014) (Fig. 7). Although all
factors necessary for silica polymorphs precipitation were available inside the sponge, it did not
precipitate due to a too low dSi concentration, indicating that the dSi concentration generated
by sponge spicules dissolution was not high enough to initiate the silica polymorphs
crystallization. The pyrite precipitation took place under euxinic conditions in the presence of
$Fe^{2+}$ ions, which were produced during OM decomposition by iron-reducing bacteria, followed
by $H_2S$ production by bacterial sulphate reduction. These distinctive geochemical conditions



led to pyrite precipitation as thefirst stable iron sulfide in the form of massive polycrystalline
textures under low Eh (Grimes et al., 2002) (Fig. 6g-i), generated inside the voids left after
spicules dissolution, simultaneously with its precipitation as octahedral crystals within the
sediment infilling the spaces between spicules, under the lower Eh than in spicules (Fig. 7). In
the studied samples, the EBSD studies revealed that marcasite is present in the form of euhedral
crystals in the sediment matrix (Fig. 6h, i), usually in areas surrounding the spicules and also
associated with pyrite building its infillings (Fig. 6h, j). Although the acid conditions generated
by OM decomposition favors the pyrite precipitation (pH ~ 6), they were simultaneously
buffered by calcite complete dissolution, which was an limiting factor for marcasite
precipitation (in pH ~4–5; Schoonen and Barnes, 1991; Murowchick and Barnes, 1987;
Benning et al., 2000). The marcasite precipitation took place via the oxidation of previously
formed pyrite (Scheiber et al., 2007) (Fig. 7). This process provided the $Fe^{2+}$, facilitate the pH
decrease to a level enabling marcasite formation, and simultaneously increased further calcite
dissolution. The presence of marcasite indicates that the redox shift occurred, probably due to
intrusion of the oxygenated porewater into the previously closed environment of the decaying
sponge body. Taking into account that marcasite precipitation is a much faster process than
pyrite formation (Schoonen and Barnes, 1991), the oxidation events were rapid and probably a
repeated process. The intrusion of porewater was possible due to ongoing process of OM
decomposition and siliceous skeleton dissolution, which created a system of holes and
passageways enabling porewater penetration. Barite ($BaSO_4$) infills the voids left after central
canal dissolution (Fig. 3b, 6g, h) and occurs in form of the patches of loosely arranged crystals
fused into one mass, usually associated with large (100–150µm) sparite crystals (Fig. 6c, e).
This distinctive barite distribution was controlled by its formation, which was governed by the
presence and distribution of OM and availability of $SO_4^{2-}$ under oxic conditions (Liguori et al.,
2016). The latter was easily available in the solution due to marcasite formation via pyrite



oxidation, which provide the $SO_4^{2-}$ (Scheiber, 2007), while the remnants of dissolved OM in
the central canal of the sponge spicule and lumps of OM surrounding the sponge skeleton acted
as an nucleation sites due to the presence an amorphous P-phase precursor, which binds the Ba
and promotes its high concentration essential for barite precipitation (Martinez-Ruiz et al.,
2020) (Fig. 7). The lack of barite within the remains of hexactinellids with rigid skeleton was
the result the small amount of OM related to the construction of these sponges, preventing barite
precipitation. After the decomposition of OM declined, the pH increased, and in a laying deeper
in the sediment sulphate reduction zone reached the level of > 7.8, enabling the calcite
reprecipitation in the form of authigenic small grains fused into homogenous coalescence/fused
texture and large sparite crystals forming cements (Fig. 6e). The distribution of the sparite
cements indicates that the presence of remnants of OM within the sediment, highlighted by
barite precipitation, were the preferential areas for newly formed sparite crystal growth due to
highly alkaline pH (~12) conditions generated by sulphate reduction bacteria, which controlled
the distinctive microtexture of the calcite. The presence of authigenic calcite grains of
distinctive microtexture only in sediment infilling the sponge (not in the surrounding sediment)
indicates that although the sponge remnants at that stage did not form a closed
microenvironment, but rather semi-closed one, the diagenetic transformations were different
from those in the surrounding sediment. This situation was caused by the presence of a
significant amount of detrital clays (up to 16% vs. 11% in sediment) only in the sediment
surrounding the sponge remnants. Moreover, the presence of clays retards the silica dissolution
(Kastner et al., 1977), leading to longer preservation of surfaces parts of sponge siliceous
skeleton that were in contact with the sediment (type 1). The clays also play an important role
as a dSi sink during diagenetic clay transformations (Fig. 6f), indicating that the dSi provided
by siliceous skeleton dissolution was absorbed by the clays, leading to a decline in the dSi
concentration, which prevented the opal-CT precipitation.



The geochemically distinct closed microenvironment forming inside the decaying sponge is
also rarely noted in opoka. In this lithology, the siliceous infillings of skeletal elements (of
mixed opal-CT and nano-α-quartz) occur occasionally, mainly in lithistids (Fig. 3j)
(Świerczewska-Gładysz, 2006). The closed microenvironment, which established around the
decaying sponge body, formed a chemical barrier for the dSi migration to the outside seabed
mud, causing in situ authigenic silica precipitation (Jurkowska and Świerczewska-Gładysz,
2020a). This distinctive microenvironment inside the dead sponge was created by microbial
decomposition of OM, which were trapped and protected from the organisms penetrating the
mud by dysoxic/anoxic conditions that stabilized inside the decaying sponge body. In samples
of bodily preserved hexactinellids with rigid skeleton, small fragments of the skeletal network
preserved as silica are extremely rare and consists of relatively large spicules (Świerczewska-
Gładysz, 2006). These sponges, due to their morphology (thin wall) did not create a close
microenvironment, and the skeletons are dissolved and infilled by ferrigenous mass of the
limonite group (Fig. 3 f-h). Thus, the presence of OM itself was not a factor that control the
silica authigenic precipitation of nano-α-quartz and lepispheric opal-CT, and only hindered the
outflow of dissolved silica from the sponge skeleton to porewater.

*5.3.3 Third factor: the impact of the mineralogical composition of the seabed mud on dSi*
*precipitation*
The mineralogical composition of the seabed mud may affect silica polymorph precipitation in
two main ways: by scavenging the dSi through clays during its diagenetic transformations or
by limiting the $Mg^{2+}$ availability, for example,. due to decreased calcite dissolution (during the
pH drop the high-Mg calcite undergoes dissolution first). In the studied rocks, clays are
significant within two lithologies: of upper Turonian-lower Coniacian and lower-middle
Campanian marls, in which their average content is 11% and 20%, respectively (Jurkowska,



2022). In both lithologies, the clays are mostly of detrital origin, but during diagenesis, they
undergo transformation, which is visible in their microtexture (Garrels and MacKenzie, 1967;
Siever and Woodford, 1973; Isson and Planavsky, 2018; Jurkowska, 2022). During these
transformations, the clays consume the dSi, Al (alkali metal cations), and alkalinity, and the
whole process is controlled by the availability of essential substrates, pH and fluid temperature
(Siever and Woodford, 1973). It has been experimentally revealed that, in seawater and
porewater under a relatively stable pH ~8, the sorption and dissolution reactions of various clay
minerals are dependent on dSi concentration. Generally, for illite/smectite, which is the most
common clay mineral, the dSi concentration at the sorption point starts at 25 ppm (250 μM),
while the kinetic sorption equilibrium for clay minerals is estimated to be approximately 60
ppm (1000 μM) (Siever and Woodford, 1973; Siever, 1991). The range of dSi concentration
(25–60 ppm) overlaps with the dSi concentration favourable for silica polymorph precipitation
(Mackenzie and Gees, 1971; Williams and Crerar, 1985), which indicates that during early
diagenesis within the seabed mud, a dynamic equilibrium between these two mineral phases
must have existed.
In the studied upper Turonian-lower Coniacian marls, the balance stabilized at at a level that
enabled the precipitation of secondary silica polymorphs only in the external part of the
siliceous sponge skeleton (while the internal spicules underwent dissolution and the voids were
infilled by different minerals) (Fig. 3c; 6a-c).In the lower-middle Campanian marls, the
precipitation of silica polymorphs took place not only as infillings of loose siliceous spicules
and skeletal networks (in the form of mixed nano-α-quartz and opal-CT (Fig. 8a-b), but also
within the sediment in the form of opal-CT lepispheres (Fig. 8c). Taking into account that
overall dSi seawater concentration was similar during the formation of both types of marls and
was at the level of opal-CT (25–60 ppm) (Siever, 1991), this indicates that the formation of
silica polymorphs (also cherts) is also dependent on the chemical equilibrium within the seabed



mud. This can lead to a situation in which, although the siliceous sponges underwent
dissolution, silica polymorphs precipitation would not be initiated. The observed mineralogical
variability in marls could also be explained by a higher overall dSi seawater concentration noted
in the Campanian (Jurkowska and Świerczewska-Gładysz, 2020a, b) compared to the
diminished dSi concentration in the Turonian-lower Coniacian. Under lower dSi concentration,
clays will be the preferential minerals that absorb dSi.

**6.** Discussion
*6.1    The role of siliceous sponges as a main source of dSi in a porewater and dSi burial from*
*a geological perspective*
In geological studies, the precipitation of cherts and authigenic silica polymorphs was treated
as a reflection of a significant amount of dSi delivered by the dissolution of siliceous sponge to
porewater, which enables the precipitation of opal-CT (Wise and de Weaver, 1974; Jeans, 1978;
Clayton, 1984, 1986; Maliva and Siever, 1989a, b; Maliva et al., 1989).The model of siliceous
chert formation presented by Siever (1986) assumes that its main mechanism was dSi
downward diffusion between seawater and porewater, which was driven by the higher
porewater dSi concentration generated by the dissolution of siliceous sponges. Unfortunately,
in many later works, although they are based on Siever (1986) mechanisms of chert formation,
the oversimplification of this mechanism led to the assumption that dSi diffusion occurred only
between the dissolving siliceous sponges and sediment mud, while the role of seawater dSi
concentration was completely ignored (Wise and de Weaver, 1974; Jeans, 1978; Clayton, 1984,
1986; Maliva and Siever, 1989a, b; Maliva et al., 1989). This simplification led to the
straightforward conclusion that the dissolution of siliceous sponges controls the porewater dSi
concentration and was the main source of dSi, governing its burial in the form of siliceous rocks.
The role of seawater as a source of dSi, even that during the Palaeozoic and Mesezoic, when its



concentration was estimated to be high (at the level of opal-CT; Siever, 1991), is interpreted as
insignificant (Maliva and Siever, 1989a, b). The results presented here indicate that although
the process of siliceous sponge skeleton dissolution was common in deposits of Cretaceous
seas, the role of the bSi (biogenic silica) inflow generated by this process in controlling dSi
concentration and as a source for silica polymorph precipitation was subordinate, while most
of the siliceous rocks were formed due to dSi downward diffusion from seawater to  porewater
(Jurkowska and Świerczewska-Gładysz, 2020a, b). As documented here, the presence of
abundant siliceous sponges (Turonian-upper Coniacian marls) which were buried did not
determine the precipitation of silica polymorphs under conditions of lower dSi concentration in
porewater (related to low dSi concentration in seawater and/or the absorption of dSi by clay
minerals), which probably occurred during this time. Similarly, preserved rigid skeletons or
loose spicules of siliceous sponges as voids infilled by iron minerals are common in the fossil
record (of Paleozoic and Mesozoic calcareous and detrital rocks) in which cherts are absent
(e.g. Rigby et al., 1995; Vodrážka, 2009; Xiao et al., 2005; Madsen et al., 2010; Reitner, 2013;
Reolid, 2014). This pattern has also been confirmed in review studies by the authors on
Palaeozoic cherts, which show that facies distribution during that time did not correlate with
siliceous sponge occurrences but is associated with dSi inflow via the volcanic-hydrothermal
activity connected with LIP (Large Igneous Provinces) (literature review in Jurkowska and
Świerczewska-Gładysz, 2024).
The presence of fossils of siliceous sponges in siliceous rock cannot be treated as a direct
argument for the biogenic origin of siliceous and carbonate-siliceous rocks, since the
fossilization process depends on many environmental factors. The studies presented here, based
on mineralogical and microtextural observations, indicate that the process of dSi migration
occurring after the  dissolution of siliceous sponge skeletons was driven by the interaction
between different sites (siliceous sponge remains, newly forming silica polymorphs of opal-





CT) controlling the dynamic equilibrium of dSi concentration, which governs the dissolution
and precipitation of silica. The observed association of occurrence of siliceous sponges within
cherts and opoka (Wise and de Weaver, 1974; Jeans, 1978; Clayton, 1984, 1986; Maliva and
Siever, 1989a, b) could be explained by the fact that basin areas characterized by significant
amount of dSi inflows were preferentially settled by silicifiers due to the availability of the main
substrate for building their skeletons. Moreover, the studies presented indicate that under high
seawater dSi concentration (as noted during the Campanian cherts formation), due to secondary
infilling of the voids left after siliceous sponge dissolution and the preservation of voids after
spicules, the fossils of siliceous sponges have higher preservation potential, which could also
affect the observations and lead to simplified conclusions.

*6.2*     *The role of seawater dSi concentration in Si circulation and burial*
The evolution of the Si cycle in Earth's history and the associated fluctuations of dSi seawater
concentrations have been presented in the classical studies by Maliva and Siever (1989a, b) and
Siever (1991). The seminal model assumes that the overall seawater dSi concentration was
relatively stable and high for most of the Palaeozoic and Mesozoic times, reaching the level of
opal-CT precipitation (250–700 µm), which is much higher than today's, which is below quartz
precipitation (< 250 µM). Our previous studies indicate that during the Cretaceous, in the
marine environment of the epicontinental European Basin, the seawater dSi concentration was
not stable and only in some areas reached the higher concentrations enabling the precipitation
of silica polymorphs (Jurkowska and Świerczewska-Gładysz, 2020a, b). Similarly, the studies
by Doering et al. (2024) indicate that during the Cretaceous, the dSi concentration of sea bottom
water was much lower than assumed in classical model (Maliva and Siever, 1989a; Siever,

701     1991).



The research conducted here indicates that the formation of cherts and the precipitation of silica
polymorph within the seabed mud are controlled by the seawater dSi concentration, which via
dSi downward diffusion to the porewater. In the studied Turonian-upper Coniacian section,
silica polymorphs did not precipitate in the seabed mud (neither as cherts nor as opoka), even
in the presence of biogenic dSi from siliceous sponges. Among the factors that could prevent
the precipitation of silica polymorphs, low seawater dSi concentration (lower than assumed for
the Cretaceous) is very probable. Even if downward dSi diffusion between seawater and
porewater occured and dSi input from the dissolution of siliceous sponges was also added to
porewater, dSi concentration was still too low to initiate the precipitation of silica polymorphs.
Unfortunately, the lack of authigenic silica polymorphs in the rocks makes it impossible to
perform analyses that could verify this hypothesis.

*6.3 The $\delta^{30}$Si signatures from fossilized sponges skeletons*
The isotopic analysis of $\delta^{30}$Si is widely used as a paleoceanographic and paleoproductivity
proxy in marine studies to constrain the Si cycle and reconstruct of past seawater dSi
concentrations (De La Rocha et al., 1997, 1998; Sutton et al., 2018; Farmer et al., 2021). In the
geological record, the materials used for e $\delta^{30}$Si measurement are siliceous rocks, mainly cherts
(e.g. Van der Boorn et al., 2007; 2010; Tatzel et al., 2015; Gao et al., 2020), and remnants of
the skeleton of silicifiers (diatoms, sponges, radiolarians) in sub-fossil and fossil states (e.g.
Egan et al., 2012; Fontorbe et al., 2017; 2020). The main limitations in using this tool for the
interpretation of paleorecords is dictated by the necessity of siliceous remnants that are built of
original biogenic opal-A, free from contaminating sources (Sutton et al., 2018). This kind of
fossil and subfossil material is very rare in the geological record (especially in rocks older than
the Paleogene) due to diagenetic changes of silica polymorphs occurring after burial. The data
presented in these studies indicate that in Cretaceous and older deposits (Palaeozoic and



Mesozoic; Jurkowska and Świerczewska-Gładysz, 2024), the siliceous remains of sponge
skeletons are built of secondary opal-CT, and this is the main common state of their preservation
(Jurkowska and Świerczewska-Gładysz, 2020a). This strongly limits the possibility of using
$\delta^{30}Si$ as a paleoceanographic proxy or for the estimation of seawater dSi concentration (Wille
et al., 2010; Hendry et al., 2012) on older paleontological material. Only single measurements
of $\delta^{30}Si$ on preserved original biogenic opal-A silicifiers older than the Eocene were performed
due to the scarcity of suitable material (Cassarino et al., 2024; Doering et al., 2024). Although
this exceptionally well-preserved material provides important data on seawater dSi
concentration during the Cretaceous, it represent only fragmented part of the whole system.
On one side, the data presented here constrain the use of $\delta^{30}Si$ of fossil material as an ad hoc
paleorecord proxy, but on the other side, the presented diagenetic models indicate that the $\delta^{30}Si$
of secondary infilled sponge skeletons could potentially carry signals indicating the dSi origin
(seawater or hydrothermal) (Fig. 4). The main questions that arises is about the share of
individual sites (which according to the diagenetic model are mainly dSi from seawater and bSi
from siliceous sponges) in the dSi of porewater from which silica polymorphs precipitation
took place, and also about the Si isotope fractionations that can occur during the process of
opal-CT precipitation followed by its later maturation within the same polymorphic phase.
In many works, the $\delta^{30}Si$ of cherts has been used for to recognize the dSi source (seawater or
hydrothermal), although it must be highlighted that this method is certainly valid for the
Precambrian and Cambrian cherts (Van der Boorn et al., 2007; 2010; Tatzel et al., 2015; Gao
et al., 2020), which were formed before siliceous sponge became an important part of the Si
cycle (Jurkowska and Świerczewska-Gładysz, 2024). During that time, due to the insignificant
role of silicifiers in Si cycle, the seawater–porewater system could be treated as closed, and the
$\delta^{30}Si$ signatures reflect the isotope fractionation occurring between the dSi source (volcano-
hydrothermal) and authigenic silica polymorph precipitation during diagenesis. Unfortunately,



in many works, cherts are used as an ad hoc paleoceanographic proxy without considering Si
isotope fractionation by silicifiers and the diagenetic impact of occurring silica polymorph
transformation. More studies based on the model presented here (and in previous works:
Jurkowska and Świerczewska-Gładysz, 2020a, b) of dSi migration and diffusion are needed to
recognize how the Si isotopes fractionate during diagenesis and what $\delta^{30}Si$ signature studies of
secondary silica polymorphs reflects.

**7.    Conclusions**

The conducted research revealed that the process of siliceous sponge dissolution within the
Cretaceous marine environment of the epicontinental basin commonly occurred in the seabed
mud and in small amount saturated the porewaters with dSi. The later occurring authigenic
silica polymorph precipitation within the seabed mud (the formation of chert and other
siliceous/carbonate-siliceous rocks) depended on environmental factors such as: overall OM
content (which, by microbial decomposition, triggered the geochemical changes facilitating
opal-CT precipitation), and the mineralogical composition of the seabed mud (in terms of the
presence of minerals that deliver essential alkalinity – carbonates – and those that scavenge the
dSi – clays). Although these agents were important, the dSi seawater concentration was the
main controlling factor of porewater dSi concentration due to constant diffusion and dSi
migration. The dynamic equilibrium of seawater-porewater dSi concentration governs the
precipitation of silica polymorphs in sediment and in voids left after spicules dissolution. High
seawater dSi concentration is recorded in the geological succession by the occurrence of cherts
and opoka, while its low concentration is reflected by the absence of these rocks, even in the
sections in where siliceous sponges are abundant and represented by remains with completely
dissolved primary skeleton. In these sections, the decaying sponge body often formed a closed
microenvironment characterized by different geochemical conditions compared to the



surrounding mud, which is visible in the geological record as the common preservation of
sponge skeletal remnants as pyrite/marcasite infillings or in the form of limonite coatings. The
skeletons of fossil sponges that are preserved as siliceous are in fact secondary infilled by
authigenic silica polymorphs (mixed nano-$\alpha$- quartz and opal-CT), which limits the usefulness
of $\delta^{30}$Si as paleoceanography proxy in geological studies but highlights its utility for the
identifying dSi origin and estimating dSi concentration.

**Data availability**

All raw data can be provided by the corresponding author upon request. The rock samples and
paleontological samples of siliceous sponges are stored at AGH University.

**Author contributions**

AJ and EŚG contributed to the idea, conceptualization and wrote the manuscript draft. AJ
conducted the mineralogical and microtextural studies of rocks and fossils and data
interpretations. EŚG worked on the taxonomy of fossils of sponge remains, and fossils sample
collections and selection. SzKF edited the manuscript, made language corrections, and prepared
samples for analysis.

**Competing interests**

The authors declare that they have no conflict of interest.

**Acknowledgements**

Many thanks go to Adam Gaweł (AGH) for valuable help with SEM-EDS analysis and to Dr.
Grzegorz Cios (AGH Academic Center for Materials and Nanotechnology) for support with
EBSD analysis.

**Financial support**

This research project is supported by program „Excellence initiative – research university" for
the AGH University.



## Literature

Alexandrowicz, S.W. and Radwan, D.: Kreda opolska - problematyka stratygraficzna i złożowa, Przegląd Geologiczny, 20, 183–188, 1973.

Benning, L.G., Wilkin, R.T. and Barnes, H.L.: Reaction pathways in the Fe-system below 100uC, Chem. Geol., 167, 25–51, 2000.

Bertolino, M., Calcinai, B., Cattaneo-Vietti, R., Cerrano, C., Lafratta, A., Pansini, M., Pica, D. and Bavestrello, G.: Stability of the sponge assemblage of Mediterranean coralligenous concretions along a millennial time span, Mar. Ecol., 35, 149–158, 2013.

Bertolino, M., Oprandi, A., Santini, C., Castellano, M., Pansini, M., Boyer, M., Bavestrello, G.: Hydrothermal waters enriched in silica promote the development of a sponge community in North Sulawesi (Indonesia), The Eur. Zoo. J., 84(1), 128–135, 2017.

Bieda, F.: Sur les spongiaires siliceux du Sénonien des environs de Cracovie, Rocznik Polskiego Towarzystwa Geologicznego, 9, 1–41, 1933.

Bromley, R.G. and Ekdale, A.A.: Trace fossil preservation in flint in the European Chalk, J. Paleontol., 58(2), 298–311, 1984.

Calvert, S.E.: Deposition and diagenesis of silica in marine sediments, in: Hsü, K.J. and Jenkyns, H.C. (Eds.), Pelagic Sediments: On Land and under the Sea. Sp. Pub. Int. Assoc. Sedimentologists, 1, 273–299, 1974.

Calvert, S.E.: Mineralogy of silica phases in deep-sea cherts and porcelanites, Mar. Mineral. Philos. Transac. Royal Soc. A, 286, 239–252, 1977.

Cassarino, L., Brylka, K., Dai, Y., Pickering, R.A., Richoz, S., Conley, D.J.: What we can learn from the oldest and first $\delta^{30}Si$ diatom taxa specific record? In Abstracts of Isotopes in Biogenic Silica (IBiS), 3. 2024.

Chu, J.W.F., Maldonado, M., Yahel, G. and Leys, S.P.: Glass sponge reefs as a silicon sink, Mar. Ecol. Prog. Ser., 441, 1–14, 2011.

Ciurej, A., Dubicka, Z. and Poberezhsky, A.: Calcareous dinoflagellate blooms during the Late Cretaceous 'greenhouse' world - a case study from western Ukraine, PeerJ, 11, e16201 http://doi.org/10.7717/peerj.16201, 2023.

Clayton, C.J.: Geochemistry of Chert Formation in Upper Cretaceous Chalk, PhD Thesis. University of London, 1984.

Clayton, C.J.: The chemical environment of flint formation in Upper Cretaceous Chalks, in: de Sieveking, G. and Hart, M.B. (Eds.), The Scientific Study of Flint and Chert, P. Fourth Int. Flint Symp. Held at Brighton Polytechnic, 10–15 April 1983. Cambridge University Press, Cambridge, 43–54, 1986.

Costa, G., Bavestrello, G., Cattaneo-Vietti, R., Dela Pierre, F.,·Lozar, F., Natalicchio, M., Violant, D., Pansini, M., Rosso, A. and Bertolino, M.: Palaeoenvironmental significance of sponge spicules in pre-Messinian crisis sediments, Northern Italy, Facies, 67, 9, 2021.



De La Rocha, C. L., Brzezinski, M. A., and DeNiro, M. J.: Fractionation of silicon isotopes by
marine diatoms during biogenic silica formation, Geochim. Cosmochim. Acta, 61(23), 5051–
841  5056, 1997.

De La Rocha, C. L., Brzezinski, M. A., DeNiro, M. J., and Shemesh, A.: Silicon isotope
composition of diatoms as an indicator of pastoceanic change, Nature, 395, 680–683, 1998.
Dias-Brito, D.: Global stratigraphy, palaeobiogeography and palaeoecology of Albian–
Maastrichtian pithonellid calcispheres: impact on Tethys configuration, Cretaceous Res., 21,
315–349, 2000.
Doering, K., Zhang, Z., Dai, Y., Dummann, W., Störling, T., Schröder-Adams, C., Richoz, S.,
Frank, M., Herrle, J., Harwood, D., Conley, D.J.: The silica cycle during the Upper
Cretaceous: the case study from Canadian Continental Shelf. In Abstracts of Isotopes in
Biogenic Silica (IBiS), 8, 2024.
Egan, K. E., Rickaby, R. E. M., Leng, M. J., Hendry, K. R., Hermoso, M., Sloane, H. J.,
Bostock, H. and Halliday, A.N.: Diatom silicon isotopes as a proxy for silicic acid utilisation:
A Southern Ocean core top calibration, Geochim. Cosmochim., 96, 174–192, 2012.
https://doi.org/10.1016/j.gca.2012.08.002, 2012.
Farmer, J. R., Hertzberg, J. E., Cardinal, D., Fietz, S., Hendry, K., Jaccard, S. L., Paytan, A.,
Rafter, P.A., Ren, H., Somes, C.J. and Sutton, J.N.: Assessment of C, N, and Si isotopes as
tracers of past ocean nutrient and carbon cycling, Global Biogeochemical Cy., 35,
e2020GB006775. https://doi. org/10.1029/2020GB006775, 2021.
Ferretti, A., Messori, F., Di Bella, M., Sabatino, G., Quartieri, S., Cavalazzi, B., Italiano, F.
and Barbieri, R.: Armoured sponge spicules from Panarea Island (Italy): Implications for their
fossil preservation, Palaeogeogr. Palaeoecol., 536, 109379,
https://doi.org/10.1016/j.palaeo.2019.109379, 2019.
Fontorbe, G., Frings, P. J., De La Rocha, C. L., Hendry, K. R., Carstensen, J., and Conley, D.
J.: Enrichment of dissolved silica in the deep Equatorial Pacific during the Eocene-Oligocene,
Paleoceanography, 32(8), 848–863, 2017.
**Fontorbe, G., Frings, P.J., De La Rocha, Ch.L., Hendry, K.R. and Conley,**
**D.J.:** Constraints on Earth System Functioning at the Paleocene-Eocene Thermal Maximum
From the Marine Silicon Cycle, *Paleoceanography and Paleoclimatology,* **35, 5**,
https://doi.org/10.1029/2020PA003873, 2020.
Gao, P., Li, S., Lash, G.G., He, Z., Xiao, X., Zhang, D. and Hao, Y.: Silicification and Si
cycling in a silica-rich ocean during the Ediacaran-Cambrian transition, Chem. Geol. 552,
119–789, 2020.
Garrels, R.M. and Mackenzie, F.T.: Origin of the Chemical Compositions of Some Springs
and Lakes (ed. Stumm W.) Equilibrium concepts in natural water systems [J], J. Am.
Chemical Soc, 222–242, 1967.
Gentric, C., Rehel, K., Dufour, A. And Sauleau, P.: Bioaccumulation of metallic trace
elements and organic pollutants in marine sponges from the South Brittany Coast, France, J.



Environ. Sci. Heal. A, 51(3), 213–219, https://doi.org/10.1080/10934529.2015.1094327,
880    2015.

Grimes, S.T., Davies, K.L., Butler, I.B., Brock, F., Edwards, D.R., Richard, Briggs, D.E.G.
and Parkes, R.J.: Fossil plants from the Eocene London clay: the use of pyrite textures to
determine the mechanism of pyritization, J. Geol. Soc. London, 159, 493–501, 2002.
Heath, G.R. and Moberly Jr., R.: Cherts from the western Pacific, leg 7, Deep Sea Drilling
Project, in: Winterer, E.L., Riedel, W.R., et al. (Eds.), Initial Rep. Deep Sea, 7. U.S.
Government Printing Office, Washington, 991–1007, 1971.
Hendry, K.R. and Robinson, L.F.: The relationship between silicon isotope fractionation in
sponges and silicic acid concentration: Modern and core-top studies of biogenic opal,
Geochim. Cosmochim. Ac., 81, 1–12, 2012.
Hurcewicz, H.: Siliceous sponges from the Upper Cretaceous of Poland, Part I, Tetraxonia,
Acta Palaeontol. Pol., 11, 15–129, 1966.
Hurcewicz, H.: Siliceous sponges from the Upper Cretaceous of Poland, Part II, Monaxonia
and Triaxonia, Acta Palaeonol. Pol., 13, 3–96, 1968.
Hurd, D.C.: Interactions of biogenic opal, sediment and seawater in the Central Equatorial
Pacific, . Geochim. Cosmochim. Ac., 37, 10, 2257–2266, 1973.
Isson, T.T. and Planavsky, N. J.: Reverse weathering as a long-term stabilizer of marine pH
and planetary climate, Nature, 560(7719), 471–475, 2018.
Jeans C.V.: Silicifications and associated clay assemblages in the cretaceous marine
sediments of southern England, Clay Miner., 13, 101–124, 1978.
Jurkowska, A.: The biotic-abiotic control of Si burial in marine carbonate systems of the pre-
Eocene Si cycle, Global Biogeochem. Cy., 36, e2021GB007079, 2022.
Jurkowska, A. and Świerczewska-Gładysz, E.: New model of Si balance in the late cretaceous
epicontinental European basin, Global Planet. Change, 186, 103–108, 2020a.
Jurkowska, A. and Świerczewska-Gładysz, E.: Evolution of late cretaceous Si cycling
reflected in the formation of siliceous nodules (flints and cherts), Global Planet. Change, 195,
103–334, 2020b.
Jurkowska, A., Świerczewska-Gładysz, E.: Opoka – a mysterious carbonate-siliceous rock: an
overview of general concepts. Geology, Geophysics & Environment, 48, 3, 257–278, 2022.
Jurkowska, A and Świerczewska-Gładysz, E. The evolution of the marine Si cycle in the
Archean-Palaeozoic - an overlooked Si source?, Earth-Science Reviews, 248, 104-629, 2024.
Jurkowska, A., Świerczewska-Gładysz, E., Bak, M. and Kowalik, S.: The role of biogenic
silica in formation of Upper Cretaceous pelagic carbonates and its palecological implications,
Cretac. Res., 93, 170–187, 2019.
Jurkowska, A., Uchman, A. and Świerczewska-Gładysz, E.: A record of sequestration of plant
material by marine burrowing animals as a new feeding strategy under oligotrophic conditions
evidenced by pyrite microtextures, Palaios, 33, 312–322, 2018.



Kastner, M., Keene, J.B. and Gieskes, J.M.: Diagenesis of siliceous oozes. I. Chemical
controls on the rate of opal-A to opal-CT transformation-an experimental study, Geochim.
Cosmochim. Ac., 41, 1041–1059, 1977.
Kędzierski, M. and Uchman, A.: Bedded chalk marls in the Opole Trough: epicratonic
deposits of the Late Cretaceous super-greenhouse episode, in: Haczewski, G. (ed.),
Guidebook for field trips accompanying 31st IAS Meeting of Sedimentology held in Kraków
on 22nd–25th of June 2015, Polish Geological Society, Kraków, 145–157, 2001.
de Kluijver, A., Nierop, K.G.J., Morganti, T.M., Bart, M.C., Slaby, B.M., Hanz, U., de Goeij,
J.M., Mienis, F. and Middelburg, J.J.: Bacterial precursors and unsaturated long-chain fatty
acids are biomarkers of North-Atlantic deep-sea demosponges, PLoS ONE, 16(1), e0241095,
https://doi.org/10.1371/journal.pone.0241095, 2021.
Kubiak, A., Pajewska-Szmyt, M., Kotula, M., Leśniewski, B., Voronkina, A., Rahimi, P.,
Falahi, S., Heimler, K., Rogoll, A., Vogt, C., Ereskovsky, A., Simon, P., Langer, E., Springer,
A., Förste, M., Charitos, A., Joseph, Y., Jesionowski, T. and Ehrlich, H.: Spongin as a Unique
3D Template for the Development of Functional Iron-Based Composites Using Biomimetic
Approach In Vitro, Mar. Drugs. 22, 21(9), 460, https://doi.org/10.3390/md21090460, 2023.
Liguori, B.T.P., de Almeida, M.G. and de Rezende, C E.: Barium and its Importance as an
Indicator of (Paleo)Productivity, Anais da Academia Brasileira de Ciências, 88, 04, 2093–
936    2103, 2016.

López-Acosta, M., Potel, C., Gallinari, M., Pérez, F.F. and Leynaert, A.: Nudibranch
predation boosts sponge silicon cycling, Sci. Rep., 20, 13(1), 1178.
https://doi/org/10.1038/s41598-023-27411-y, 2023.
Mackenzie F.T. and Gees G.: Quartz: Synthesis at Earth-Surface Conditions, Science, 173,
533–535, 1971.
Madsen, H.B., Stemmerik, L. and Surlyk, F.: Diagenesis of silica-rich mound-bedded chalk,
the Coniacian Arnager Limestone, Denmark, Sediment. Geol., 223, 1–2, 2010.
Maliva, R.G., Knoll, A.H. and Siever, R.: Secular change in chert distribution: a reflection of
evolving biological participation in the silica cycle, Palaios 4, 519–532 1989.
Maliva, R.G. and Siever, R.: Nodular chert formation in carbonate rock, J. Geol. 97 (4), 421–
433, 1989a.
Maliva, R.G. and Siever, R.: Chertification histories of some Late Mesozoic and Middle
Palaeozoic platform carbonates, Sedimentology, 36, 907–926, 1989b.
Martinez-Ruiz, F., Paytan, A., Gonzalez-Muñoz, M., Jroundi, F., Abad, M., Lam, P., Horner,
T. and Kastner, M.: Barite precipitation on suspended organic matter in the mesopelagic zone,
Front. Earth. Sci., 8, 567714, https:/doi.org/10.3389/feart.2020.567714, 2020.
Meister, P., Herda, G., Petrishcheva, E., Gier, S., Dickens, G.R., Bauer, C. and Liu, B.:
Microbial Alkalinity Production and Silicate Alteration in Methane Charged Marine
Sediments: Implications for Porewater Chemistry and Diagenetic Carbonate Formation, Front.
Earth Sci., 9, 756591, https://doi.org10.3389/feart.2021.756591, 2022.



Murowchick, J.B. and Barnes, H.L.: Effects of temperature and degree of supersaturation on pyrite morphology, Am. Mineral., 72, 1241–1250, 1987.

Płachno, B., Jurkowska, A., Pacyna G., Worobiec, E., Gedl, P. and Świerczewska-Gładysz, E.: Late Turonian plant assemblage from Opole (southern Poland): new data on Late Cretaceous vegetation of the northern part of European Province in the light of palaeoenvironmental studies, P. Geologist Assoc., 129, 159–170, https://doi.org/10.1016/j.pgeola.2018.01.008, 2018.

Pożaryska, K.: The sedimentological problems of Upper Maastrichtian and danian of Puławy Environment (Middle Vistula), Biuletyn Państwowego Instytutu Geologicznego, 81, 1–104, 1952.

Reolid, M.: Pyritized radiolarians and siliceous sponges from oxygen-restricted deposits (Lower Toarcian, Jurassic), Facies, 60, 789–799, https://doi.org/10.1007/s10347-014-0404-6, 2014.

Reitner, J.: A rare new demosponge from the Solnhofen Lithographic Limestone (Upper Jurassic, Bavaria, Germany), Doc. naturae, 192.4, 359-371, 2013.

Rigby, J. K. and Craig R. C. Demosponges and Hexactinellid Sponges from the Lower Devonian Ross Formation of West-Central Tennessee, J. Paleontol. 69, 2, 211–232, http://www.jstor.org/stable/1306253, 1995.

Robert, F. and Chaussidon, M.: A palaeotemperature curve for the Precambrian oceans based on silicon isotopes in cherts, Nature, 443, 969–972, 2006.

Rützler, K. and Macintyre, I.: Siliceous sponge spicules in coral reef sediments, Mar. Biol., 49, 147–159, 1978.

Schieber, J.: Oxidation of detrital pyrite as a cause for marcasite formation in marine lag deposits from the Devonian of the eastern US, Deep-Sea Res. II, 54, 1312–1326, 2007.

Schoonen, M.A.A. and Barnes, H.L.: Reactions forming pyrite and marcasite from solution. I. Nucleation of FeS2 below 100u C, Geochim. Cosmochim. Ac., 55, 1495–1504, 1991.

Siever R.: Oceanic silica geochemistry and nodular chert formation, Geol. Soc. Am., Abstract Progress, 18, 750, 1986.

Siever, R.: Silica in the oceans: Biological-geochemical interplay, in: Schneider, S. H. and Boston, P.J. (Eds.): Scientists on Gaia. MIT Press, Cambridge, 287–295, 1991.

Siever, R. and Woodford, N.: Sorption of silica by clay minerals, Geochim. Cosmochim. Ac., 37, 1851–1880, 1973.

Stanley, M.S., Ries, J.B. and Hardie, L.A.: Seawater chemistry, coccolithophore population growth, and the origin of Cretaceous chalk, Geology, 33, 593–596, 2005.

Stratmann, T., Simon-Lledó, E., Morganti, T.M., de Kluijver, A., Vedenin, A. and Purser, A.: Habitat types and megabenthos composition from three sponge-dominated high-Arctic seamounts, Sci Rep., 29,12(1),20610, https://doi.org/10.1038/s41598-022-25240-z, 2022.



Sujkowski, Z.: Petrografia kredy Polski. Kreda z głębokiego wiercenia w Lublinie w
porównaniu z kredą niektórych innych obszarów Polski, Sprawozdania Państwowego
Instytutu Geologicznego, 6, 485–628, 1931.
Sutton, J., Andre, L., Cardinal, D., Conley, D.J., de Souza, G.F., Dean, J., Dodd, J., Ehlert, C.,
Ellwood, M.J., Frings, P.J., Grasse, P., Hendry, K., Leng, M.J., Michalopoulos, P., Panizzo,
V.N. and Swann, G.E.A.: A review of the stable isotope bio-geochemistry of the global
silicon cycle and its associated trace elements, Front. Earth Sci., 5, 1–24, 2018.
Świerczewska-Gładysz, E.: Late Cretaceous siliceous sponges from the Middle Vistula River
Valley (Central Poland) and their palaeoecological significance, Ann. Soc. Geol. Pol., 76,
227–296, 2006.
Świerczewska-Gładysz, E.: Late Turonian and early Coniacian ventriculitid sponges
(Lychniscosida) from Opole Trough (southern Poland) and their palaeoecological
significance, Ann. Soc. Geol. Pol., 82, 201–224, 2012a.
Świerczewska-Gładysz, E.: Hexactinellid sponge assemblages across the Campanian-
Maastrichtian boundary in the Middle Vistula River section, central Poland, Acta
Geol. Pol., 62, 561–580, 2012b.
Świerczewska-Gładysz, E.: Early Campanian (Late Cretaceous) Pleromidae and
Isoraphiniidae (lithistid Demospongiae) from the Łódź-Miechów Synclinorium (central and
southern Poland): new data and taxonomic revision, Pap. Palaeontol., 2, 189–321, 2016.
Świerczewska-Gładysz, E. and Jurkowska, A.: Occurrence and paleoecological significance
of lyssacinosid sponges in the Upper Cretaceous deposits of southern Poland, Facies, 59, 763–
1015 777, 2013.

Świerczewska-Gładysz, E., Jurkowska, A. and Niedźwiedzki, R.: New data about the
Turonian–Coniacian sponge assemblage from Central Europe, Cretaceous Res., 94, 229–258,
1018 2019.

Tatzel, M., von Blanckenburg, F., Oelze, M., Schuessler, J.A. and Bohrmann, G.: The silicon
isotope record of early silica diagenesis, Earth Planet. Sci. Lett. 428, 293–303, 2015.
Van den Boorn, S.H.J.M., van Bergen, M.J., Nijman, W. and Vroon, P.Z.: Dual role of
seawater and hydrothermal fluids in Early Archean chert formation: evidence from silicon
isotopes, Geology 35, 939–942, 2007.
Van den Boorn, S.H.J.M., van Bergen, M.J., Vroon, P.Z., de Vries, S.T. and Nijman, W.:
Silicon isotope and trace element constraints on the origin of ~3.5 Ga cherts: implications for
early Archaean marine environments, Geochim. Cosmochim. Acta 74, 1077–1103, 2010.
Van Dijk, I., de Nooijer, L.J., Hart, M.B. amd Reichart, G-J.: The long-term impact of
magnesium in seawater on foraminiferal mineralogy: mechanism and consequences, Global
Biogeochem. Cy., 30, 438–446, https://doi.org/10.1002/2015GB005241, 2016.
Vodrážka, R.: A new method for the extraction of macrofossils from calcareous rocks using
sulphuric acid, Palaeontology, 52, 187–192, 2009.]




Von Rad, U. and Rösch, H.: Petrology and diagenesis of deep sea cherts from the Central Atlantic, in: Hsü, K.J. and Jenkins, H.C. (Eds.), Pelagic Sediments: On Land and under the Sea, Sp. Publ. Int. Assoc. Sedimentologists, 1, 327–347, 1974.

Walaszczyk, I.: Inoceramid stratigraphy of the Turonian and Coniacian strata in the environs of Opole (southern Poland), Acta Geol. Pol., 38, 51–61, 1988.

Walaszczyk, I.: Integrated stratigraphy of the Campanian–Maastrichtian boundary succession of the Middle Vistula River (central Poland) section; introduction. Acta Geol. Pol., 62, 4, 485–493, https://doi.org/10.2478/v10263-012-0027-6, 2012.

Walaszczyk, I., Dubicka, Z., Olszewska-Nejbert, D. and Remin, Z.: Integrated biostratigraphy of the Santonian through Maastrichtian (Upper Cretaceous) of extra-Carpathian Poland. Acta Geol. Pol. 66, 3, 313–350, https://doi.org/10.1515/agp-2016-0016, 2016.

Williams, L.A. and Crerar, D.A.: Silica diagenesis, II. General mechanisms, J. Sediment. Petrol., 55 (3), 312–321, 1985.

Wille, M., Sutton, J., Ellwood, M. J., Sambridge, M., Maher, W., Eggins, S. and Kelly, M.: Silicon isotopic fractionation in marine sponges: A new model for understanding silicon isotopic variations in sponges, Earth Planet. Sc. Lett. 292, 3–4, 281–289, https://doi.org/10.1016/j.epsl.2010.01.036, 2010.

Wise, S.W. and de Weaver, F.M.: Chertification of oceanic sediments, in: Hsü, K.J. and Jenkyns, H.C. (Eds.), Pelagic Sediments: On Land and under the Sea, Sp. Pub. Int. Assoc. Sedimentologists, 1, 301–326, 1974.

Xiao, S., Hu, J., Yuan, X., Parsley, R.L. and Ruiji Cao R.: Articulated sponges from the Lower Cambrian Hetang Formation in southern Anhui, South China: their age and implications for the early evolution of sponges, Palaeogeogr. Palaeocl., 220, 1–2, 89-117, 2005.

Zjilstra, H.J.P.: Early diagenetic silica precipitation, in relation to redox boundaries and bacterial metabolism in Late Cretaceous Chalk of the Maastrichtian type locality, Geol. Mijnb., 66, 343–355, 1987.

Zjilstra, J.J.P.: Sedimentology of the Late Cretaceous and Early Tertiary (Ttuffaceous) Chalk of Northwest Europe, Geol. Ultraiect., 119, 1–192, 1994.

**Figure caption:**



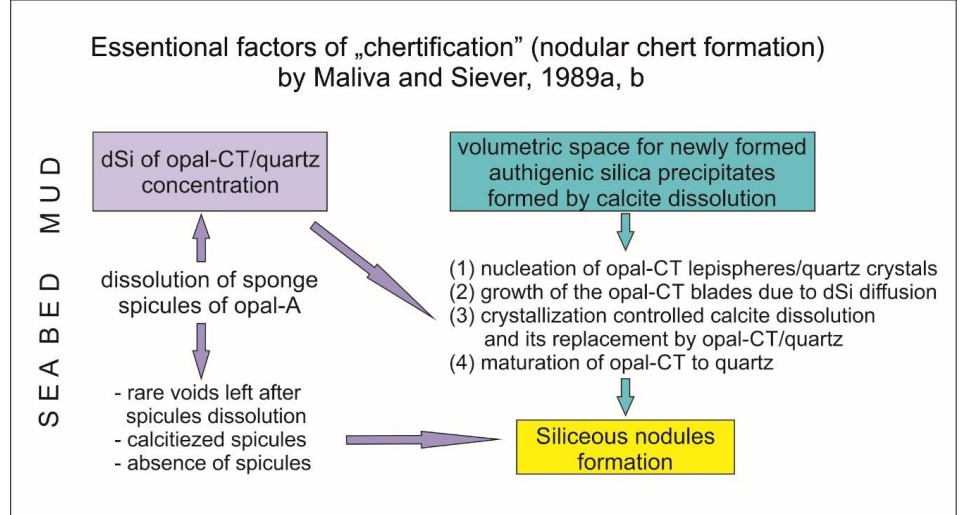

**Figure 1.** The main assumptions of the classical "chertification" model (Maliva et al., 1989), and essential factors and diagenetic stages of chert formation.

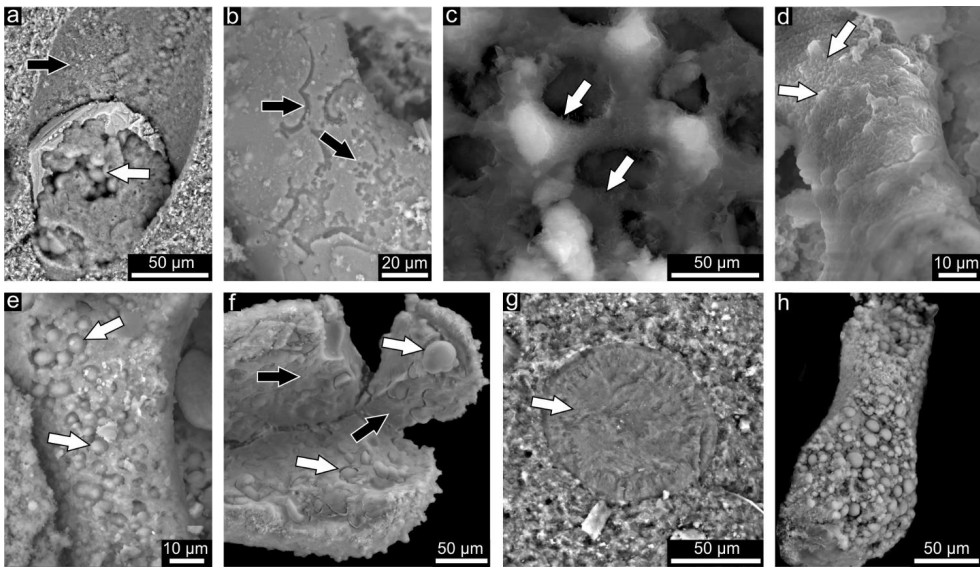

**Figure 2.** The mineralogy and microtexture of siliceous remnants of sponge loose spicules and rigid skeletons. **(a)** The siliceous infilling of the spicules with a visible external smooth layer of nano-α-quartz (black arrow) and internal infilling of lepispheres of opal-CT (white arrow); sample Wierzb.1_20 ch. **(b)** Dissolution remarks formed as cavernous pattern, rounded and platy pits (white arrow) visible on external quarzitic layer of sponge spicule; sample RajN.1_17 op. **(c)** A dense layer of opal-CT covering (white arrow) the external nano-α-quartz layer of



sponge skeleton; sample Wierzb.1_23 op. **(d)** Mixed opal-CT clayey/early forms of embryonic opal-CT layer (white arrow) covering the external surface of sponge spicule; sample RajN.1_13op. **(e)** Lepispheres of opal-CT (white arrow) infilling the dissolution rounded remarks; sample Piotr.1_7 op. **(f)** The lepispheres of opal-CT (white arrow) cemented by porous silica (black arrow) infilling the spicule; sample Wierzb.1_16 op. **(g)** A homogenous dense mass of nano-α-quartz mixed with opal-CT infilling the sponge spicule (white arrow); sample Wierzb.1_9 ch-fl. **(h)** An internal infilling of sponges spicule composed of lepispheres of opal-CT cemented by porous silica; the external layer of nano-α-quartz is not preserved; sample Jeż.2_2 ch.

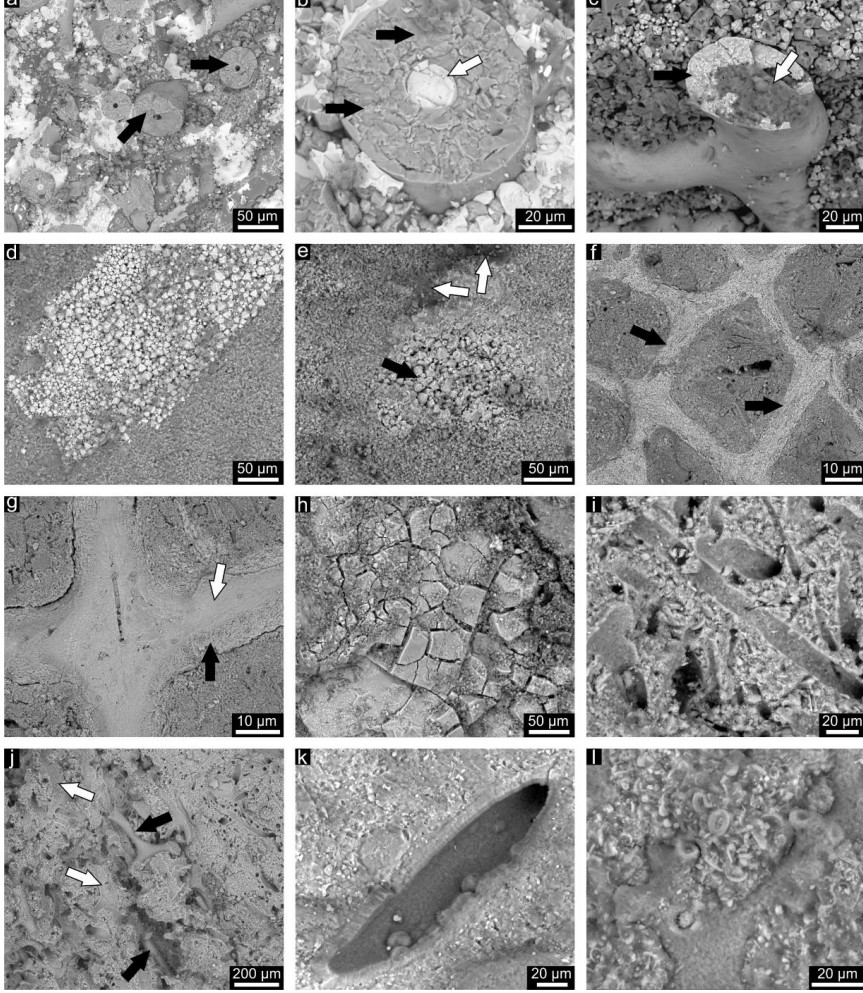

**Figure 3.** The state of preservation of siliceous sponges under the SEM. **(a, b)** The pyrite and marcasite infillings of the sponge spicules (black arrow) with visible polycrystalline texture of pyrite and smooth texture on the external part of the spicule; the white arrow indicates barite;





non-rigid demosponge, upper Turonian marls, sample Fol.7_002. **(c)** The siliceous skeletal network of lithistid sponge with nano-α-quartz and opal-CT (white arrow) and pyrite with marcasite infilling; upper Turonian marl, sample Fol.17_001. **(d)** The pyrite crystals outlining the voids left after the dissolution of hexactinellid sponge spicules; Turonian limestone, sample Fol.2_001. **(e)** The pyrite crystals with oxidation remarks (black arrow) and associated lumps of OM (white arrow); hexactinellid sponge, upper Turonian limestone, sample Fol.2_001. **(f)** The mixed pyrite and ferrigenous coatings outlining the previous siliceous skeletal network of hexactinellid sponge (black arrows); lower Maastrichtian opoka, sample Dziur.14_001. **(g)** The ferruginous coatings forming a smooth texture inside (white arrow) and with a cavernous pattern on the outside of the hexactinellid sponge spicule (white arrow); upper Campanian opoka, sample Piotr.11_001. **(h)** Ferruginous coatings as blocky microtexture; hexactinellid sponge, middle Campanian opoka, sample Rzeź.19_001. **(i)** The voids left after spicules dissolution, disperse spicules of non-rigid sponges, Campanian opoka, sample Rzeź.19_001. (j) The voids left after rigid skeleton dissolution of lithistid sponge (white arrow) with some siliceous infilling of spicules (black arrow) and opoka with voids left after loose spicules dissolution of non-rigid sponges (top-right); upper Campanian opoka, sample Piotr.12_011. (k) The voids left after spicules dissolution of non-rigid sponges within chert covered by a single layer of opal-CT; middle Campanian chert, sample Rzeź.9_001. (l) The siliceous infilling of the spicules of lithistid sponge within chert; lower Campanian chert, samples Pniaki.24_002.



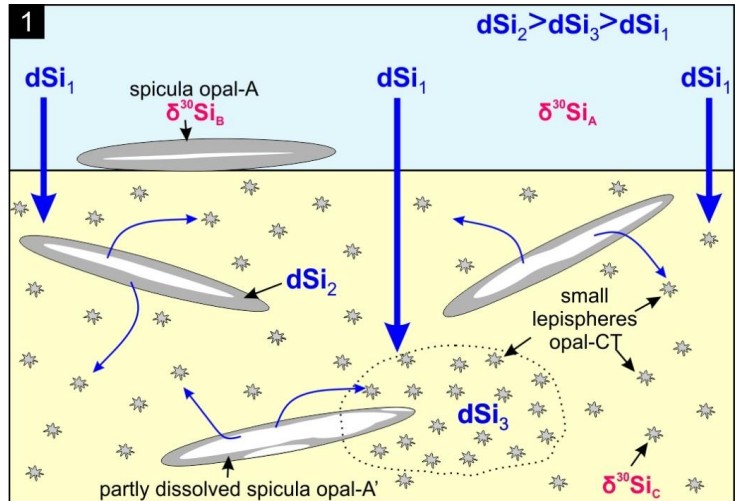

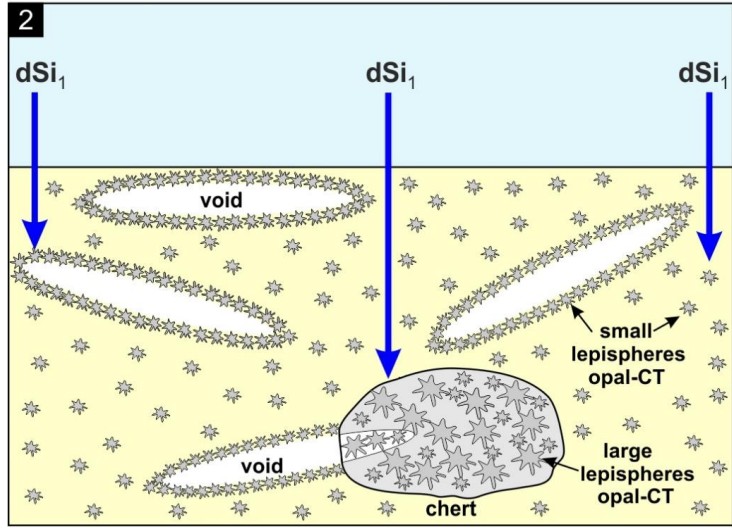

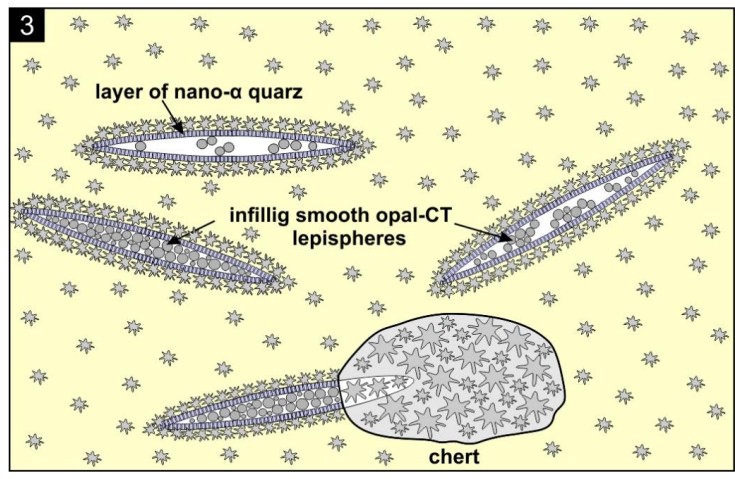





**Figure 4.** The stages of siliceous sponge dissolution followed by secondary silica polymorph precipitation (opal-CT) under high seawater dSi concentration and the presence of essential factors for dSi precipitation. Description: text chapter 5.2. The $\delta^{30}Si_A$- the Si isotope signature in a seawater of probably values of hydrothermal (-0.2 to -0.7) (Robert and Chaussidon, 2006) or seawater (0.7 to 2.2) (Sutton et al., 2018); $\delta^{30}Si_B$ – the Si isotope signature of sponge spicules (0 to -6) (Sutton et al., 2018) with more negative values with increasing seawater dSi concentration (Sutton et al., 2010); $\delta^{30}Si_c$-?

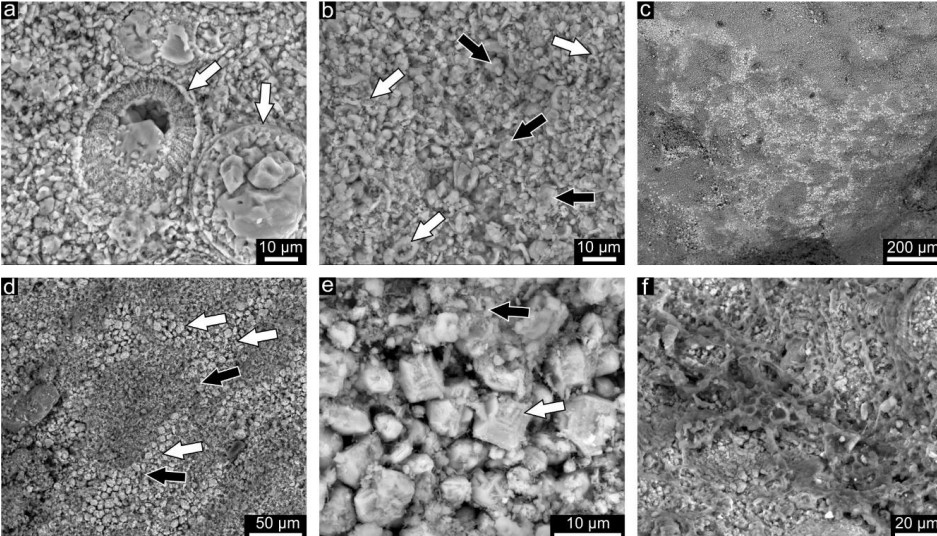

**Figure 5**. Turonian limestone and the state of preservation of siliceous sponges from these deposits. **(a)** The common calcispheres of pithonellid assemblages documented in sediment matrix (white arrow); sample Fol.2_001. **(b)** The sediment matrix of limestone with visible allomicritic calcite grains of coccoliths and its fragments (white arrow) and rare authigenic grains (black arrow); sample Fol.2_002. **(c-e)** The sponges skeletons are preserved as mixed euhedral pyrite (white arrow) with subordinate ferrigenous (limonite group) coatings (black arrow) outlining the previous siliceous skeleton; sample: Fol.3_001. **(f)** The lumps of OM associated only with sponge skeleton; sample Fol.2_001.




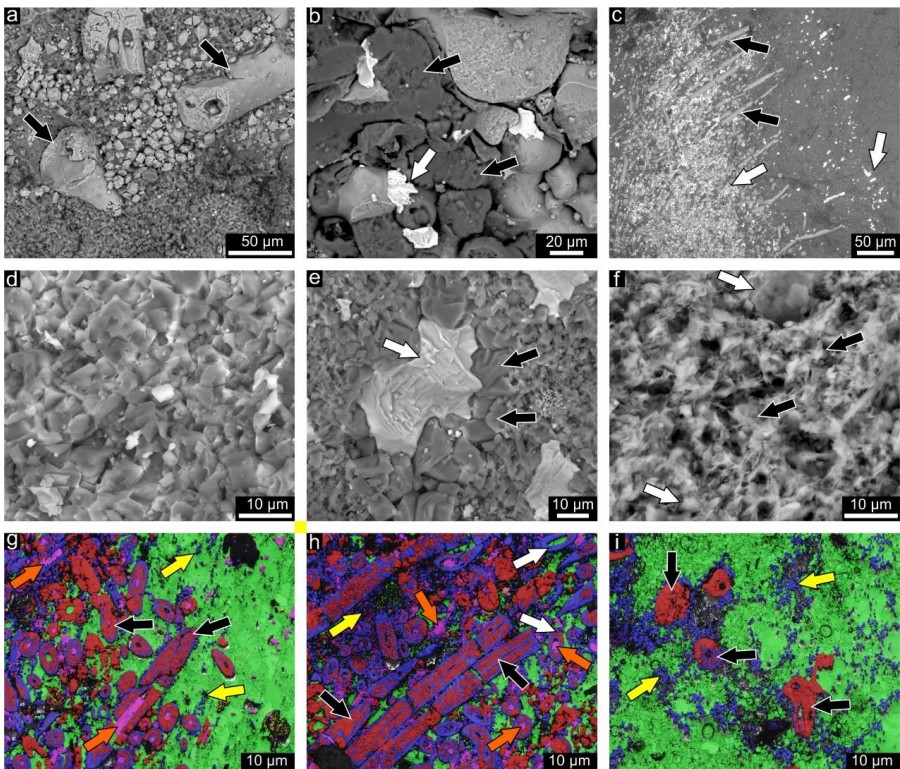

**Figure 6**. The state of preservation of siliceous sponges in upper Turonian–lower Coniacian marls. **(a)** The sponge spicules infilled by the polycrystalline pyrite with marcasite (black arrow) of hexactinellid sponge; sample Fol.4_001. **(b)** The siliceous skeleton of mixed opal-CT and nano-α-quartz (black arrow) with rare barite (white arrow) of lithistid demosponge (type 1 of preservation); sample Fol.17_001. **(c)** The spicules preserved as massive polycrystalline pyrite with marcasite (black arrow) and distribution of barite (white arrow) in non-rigid demosponge (type 2 of preservation); sample Fol.7_001. **(d)** The coalescence fused structure of authigenic calcite grains infilling the spaces between spicules of non-rigid demosponge; sample Fol.7_001. **(e)** The barite (white arrow) surrounded by large sparite crystals (black arrow) occurring in deposits infilling the spaces between spicules of non-rigid demosponges; sample Fol.7_001. **(f)** The sediment surrounding the hexactinellid sponge fossils with visible diagenetically transformed clays (black arrow) and rare authigenic calcite grains (white arrow); sample Fol.4_001. **(g-i)** The EBSD analysis (green – calcite, blue – marcasite, red – pyrite, purple – barite); the spicules infilled by the pyrite and marcasite (black arrow) and marcasite within the sediment (yellow arrow), barite infilling the void left after the dissolution of the spicules area around the central canal (orange arrow); non-rigid demosponge, sample Fol.7_001 (g, h); hexactinellid sponge, sample Fol.4_001 (i).



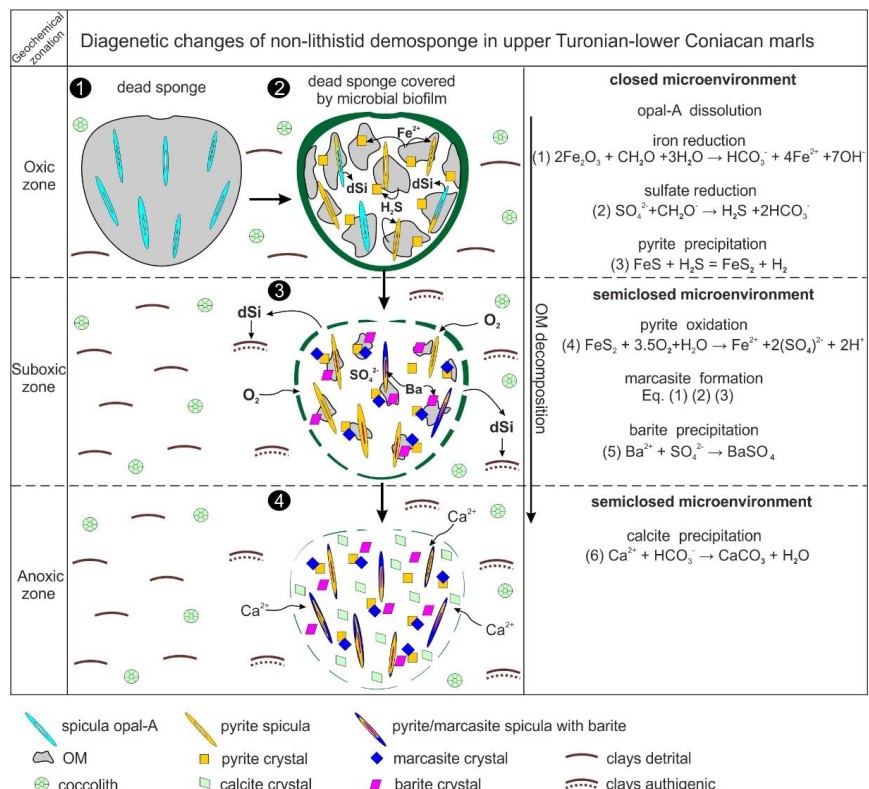

**Figure 7.** The diagenetic changes of non-lithistid demosponges in upper Turonian-lower Coniacian marls. Description: text chapter 5.3.2.

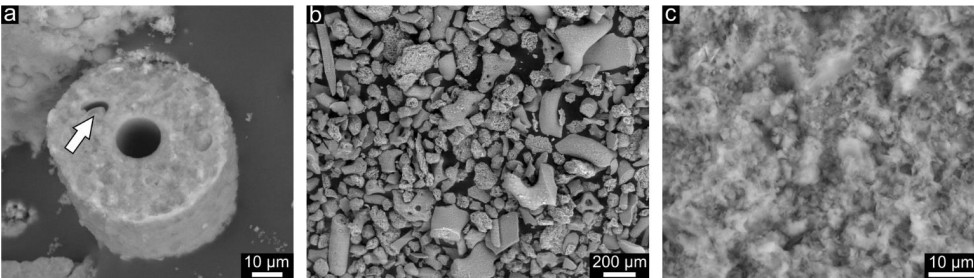

**Figure 8.** The state of preservation of siliceous sponges in lower-middle Campanian Marls of MS. **(a, b)** The siliceous of mixed nano-α-quartz and opal-CT infilling of the sponge spicules with visible dissolution remarks (white arrow); sample. B.Wlk. 1_4_001. **(c)** The opal-CT lepispheres in the sediment matrix of marls; sample Jeż.2_5_002.



| Section | Age | Lithology | Literature |
|---|---|---|---|
| Opole area: Folwark section | upper Turonian-lower Coniacian | marls, limestone | Alexandrowicz and Radwan, 1973; Walaszczyk, 1992; Kędzierski and Uchman, 2001; Świerczewska-Gładysz, 2012 |
| MS: Jeżówka 2, Wierzbica, Biała Wielka, Pniaki, Rzeżuśnia | lower-middle Campanian | marls | Jurkowska, 2016; Jurkowska and Świerczewska-Gładysz, 2020b; Jurkowska, 2022 |
| MS Jeżówka 2, Wierzbica, Biała Wielka, Rzeuśnia, Pniaki | lower-middle Campanian | opoka intercalated with cherts | Jurkowska, 2016; Jurkowska and Świerczewska-Gładysz, 2020b; Jurkowska, 2022 |
| MVR: Piotrawin, Raj N, Dziurków, Pawłowice Cm., Dorotka | middle Campanian-lower Maastrichtian | opoka | Walaszczyk, 2012; Walaszczyk et al., 2016; Jurkowska and Świerczewska-Gładysz, 2020a |

1165

1166  **Table 1.** Studied sections, lithology and literature.

1167

1168

1169