# Peer review of "The role of siliceous sponges in pre-Eocene marine Si cycle from the perspective rock"

_EGUsphere, 2024_

## Author Comment (AC2)

Below we present a reply to the Anonymous Reviewer's comments

Thank you for your review, which improved our manuscript.

General comments

Reviewer: The manuscript by Jurkowska et al. addresses Si burial in the oceans prior to the Ecocene. They performed some detailed microtextural and mineralogical analysis in Cretaceous siliceous rocks

*Authors: Our studies were based mainly on the paleontological material of fossils of siliceous sponge spicules/skeletons, focusing on mineralogical and microtextural analysis, while the rock samples were considered of secondary importance. Based on this comment, we decided that the title of the manuscript could be misleading, and it will changed it into: The role of siliceous sponges in the pre-Eocene marine Si cycle: a perspective from fossils sponge mineralogy.*

Reviewer: (…) then they concluded a relatively closed system for the decay and dissolution of siliceous sponges, thus a negligible role of sponges in regulating dissolved silica (dSi) in porewater.

*Authors: We disagree with the statement provided. We analyze the main factors controlling the dissolution of the siliceous skeleton during diagenesis (as detailed in Chapter 3). We describe the siliceous sponge skeleton and sediment as an open geochemical system working between: the decaying sponge body, seawater and sediment (as detailed in Chapter 5.2) In terms of dSi dynamic diffusion between these sites as the main process regulating the dSi concentration in porewater. We also describe the closed system of decaying sponge body but in a rocks in which the cherts does not occur, so it does have an negligible impact on Si circulation (Chapter 5.3). In Chapter 5.2, we clearly indicate that the driving force regulating the dSi concentration in porewater was a dynamic balance achieved through dSi diffusion between three sites: seawater/porewater, sponge skeleton, and newly forming cherts. This diffusion was driven by constant inflow of dSi from seawater. This system was much more complex, and the statement made by the Reviewer is misleading and does not accurately reflect what we present in our studies. The closed microenvironment of the decaying sponge body plays a decisive role only in terms of iron sulphide precipitation in marly deposits (Chapters 5.3 and 5.4). We provide additional arguments and explanations to support our idea of the negligible role of siliceous sponges in regulating dSi concentration in porewater (Chapter 6.2).*

Reviewer: Jurkowska et al. question the current concepts regarding chert formation and present their study based on mineralogical and microtextural analysis of rock and sponge remains.

*Authors: In this manuscript, we do not discuss the previous and current concepts of chert and flint formation models. We have presented and discussed a new model for the origin of siliceous nodules in previous articles (Jurkowska and Świerczewska-Gładysz, 2020a, b; Jurkowska, 2022; Jurkowska and Świerczewska-Gładysz, 2024), where we present all the data, the model, and discuss it in the context of previous research. This is not the subject of the studies presented here, where we discuss in detail the role of siliceous sponges dissolution in the supply of dSi to porewater and controlling of dSi concentration. This process (Fig. 1) is one of the stages of chert formation.*

Reviewer: In addition, they question studies using d30Si.

*Authors: We do not agree with that statement, which is oversimplified and leds to incorrect conclusions. Throughout the manuscript, we do not question the overall $\delta^{30}Si$ studies, which are very useful in many modern studies and for relatively young fossil or paleontological material composed of primary silica polymorphs. We are familiar with that literature and we collaborate with researchers who work on this material. In the presented manuscript we discuss of the use of $\delta^{30}Si$ in **studies of fossil material composed not of primary silica polymorphs, but of secondary polymorphs such as opal-CT and nano-α-quartz, and this is clearly stated throughout the manuscript (e.g conclusion: "**The skeletons of fossil sponges that are preserved as siliceous are in fact secondary infilled by authigenic silica polymorphs (mixed nano-α- quartz and opal-CT), which limits the usefulness of $\delta^{30}Si$ as paleoceanography proxy in geological studies but highlights its utility for the identifying dSi origin and estimating dSi concentration". We emphasize that in Cretaceous material (and older), where in most sections the fossils of silicifiers are preserved as secondary nano-α-quartz and opal-CT, the $\delta^{30}Si$ signals need to be treated with caution, as they are not preserved primary geochemical signals. The preservation of siliceous fossil material as opal-A in Cretaceous and older deposits is very rare, but it has been documented (e.g. Doering et al., 2024). To ensure that our assumption is interpreted correctly, we reviewed the entire text and added the appropriate clarification.*

Reviewer: The authors are applying the concept that prior to the Eocene the biological Si cycle was dominated by sponges and radiolaria, and that diatoms became dominant at the beginning of the Eocene reducing dSi in the surface oceans to the low levels observed today (Siever 1991). This assumption has important implications for their model of Si burial in the pre-Ecocene time period. However, the literature on molecular clocks suggest that diatoms evolved over 200 Ma ago (Nakov et al. 2018, New Phytologist 219:462-473), although this result has been largely ignored because of the lack of diatoms in the geologic fossil record. Diatoms likely had a major impact on the Si cycle earlier than the Cenozoic (Conley et al. 2017, doi: 10.3389/fmars.2017.00397).

*Authors: Generally, the paleontological record of fossil preservation is a basis for recognizing the evolution of organisms in Earth's history in geological studies, while the evidence such as molecular clocks is indirect. However, in our previous studies, we checked whether diatoms are present in the Campanian-Maastrichtian succession of MS and MVR, and only single fossils were found. We agree that diatoms evolved over 200 Ma, but their lack in geological record indicate that they were not abundant (and their distribution was probably restricted to polar regions) and played a marginal role in the Si cycle before the Eocene. We are familiar with the article by Conley et al.. (2017), and part of the discussion regarding it was published in our article, i.e.: Jurkowska and Świerczewska-Gładysz, 2024. Conley et al. (2017) agreed with the statement that during the Cretaceous, the dSi concentration was much higher that today (though not as high as indicated by Siever, 1991), and this implies that the lack of diatom fossils cannot be attributed to taphonomic effect, especially when the other small siliceous organisms/part of skeletons (radiolarians, sponge microskleres) are preserved in paleontological record.*

Reviewer: While it is true that there are more diatoms in the geologic record during the Cenozoic, there is more evidence that there were many species of diatoms in the Upper Cretaceous (review by Brylka et al. 2024, Marine Micropaleontology 190:102371).

*Authors: We agree, but still were they quantitatively abundant? Even if they were, what role did they play in dSi circulation if the volcanic and hydrothermal dSi sources connected with Large Igneous Provinces (LIP) and plate tectonics where much higher than today? The main subject of this article was to trace the role of siliceous sponges in Si burial and chert formation;*

*we did not detect any other silicifiers in significant amounts in the paleontological record. It is beyond the scope of this ms to discuss the role of diatoms, as we concentrate on sponge material. In the studied section, under the geochemical conditions described (Jurkowska, 2022) within the seabed mud, the diatoms would be preserved. Therefore, their absence suggests that they were not present.*

Reviewer: Diatoms dissolve in sediments with temperatures over 30oC and were likely dissolved as they were buried.

*Authors: They would not have dissolved in the Cretaceous seawater under high dSi concentrations (above the quartz concentration) and under the geochemical conditions of the carbonate seabed mud in the Upper Cretaceous European Basin (Jurkowska et al., 2020a). Even smaller siliceous skeletal elements (radiolarians and microscleres) are present in the paleontological record. If the postulated temperatures were achieved in the studied sediments, other diagenetic transformations of silica polymorphs (opal-CT) presented in the studied rocks would occur and would be visible in the rock microtexture. However, no signs of such transformation have been detected in the studied rocks (Jurkowska and Świerczewska-Gładysz, 2020; Jurkowska, 2022). If the process of silica dissolution had occurred, other elements of the siliceous skeleton would also show signs of it, while in the studied samples, the precipitation of silica polymorphs was the last process noted from sediment stratigraphy. Moreover, the process of silica dissolution during late diagenesis is not initiated solely by the temperature; other factors are essential (Kastner et al. 1977).*

Reviewer: Although the Lower Cretaceous record is limited, the global distribution of study sites and the diversity of the oldest diatoms point towards earlier dispersal and diversification events. Indirect evidence for the earlier evolution is also provided through molecular clocks. The taxonomic richness and geographical spread of these diatom communities suggest prior evolutionary events. Altogether the distribution of diatom deposits and their diversity in the Lower Cretaceous on both hemispheres, suggests that the earliest diatoms are yet to be discovered. Diatoms likely had been diversifying before 120 Ma to evolve into separate lineages and dispersed, adapted, and prevailed in new environments where they further diversified. Eventually, only a small proportion of these communities were preserved in sediments. Therefore, the assumption used by Jurkowska et al. that diatoms do not have a role in removing dSi from the oceans prior to the Eocene is likely incorrect.

*Authors: This comment does not pertain to the manuscript under discussion. In the studies presented here, we do not discuss the role of diatoms. Therefore, please specify in which of our papers the statement has been published?*

Reviewer: Jurkowska et al. also question the "Usefulness of stable isotopic studies of d30Si in geological studies of fossils." in lines 27-39.

*Authors: Lines 27-39: "The data presented here about the diagenesis of siliceous sponges skeletons opens the discussion on the usefulness of stable isotopic studies of $\delta^{30}Si$ in geological studies of fossils of silicifiers preserved as secondary silica polymorphs (opal-CT)." As we have written here, we are not questioning the usefulness of this method; we are opening a discussion about using it for the study of fossils of silicifiers preserved as secondary silica polymorphs (opal-CT). There is a significant difference between the Reviewer's interpretation and what we actually wrote in the ms.*

Reviewer: The transformation from opal-a to opal-CT is a dissolution precipitation reaction and these diagenesis reactions are well known to fractionate Si isotopes. This does not invalidate all uses of Si isotopes in geologic studies.

*Authors: We agree, and this is what we assume in the Chapter 6.3, "The $\delta^{30}Si$ signatures from fossilized sponges skeletons," in the following sentence: "The main limitations in using this tool ($\delta^{30}Si$ ) for the interpretation of paleorecords is dictated by the necessity of siliceous remnants that are built of original biogenic opal-A, free from contaminating sources (Sutton et al., 2018)."*

Reviewer: What should be done is that the material should be measured by x-ray diffraction to determine if the material is opal-a or has been transformed to opal-CT. The literature using Si isotopes from sponge spicules that are opal-a shows that during the last 90 Ma dSi bottom water dSi concentrations in different parts of the ocean range from 0-70 um (Conley et al. 2027; Dai et al. 2022; Doering et al. 2024; Fontorbe et al. 2016, 2017, 2020; Störling et al. 2024).

*Authors: Exactly! We cite those publications in the ms and highlight that the $\delta^{30}Si$ analyses were performed on primary silica material (please refer to Chapter 6.3).*

Reviewer: The idea that chert formation is governed by "volcano-hydrothermal Si sources" is primarily supported by Jurkowska's publications. In the GBC 2021 paper by Jurkowska suggest that the dSi is released into seawater by volcano-hydrothermal sources and then transported by o cean currents into the basin.

*Authors: As we mentioned before, the model of Si cycle is not the primary subject of this article. The process of siliceous sponge dissolution is discussed in light of the previously presented model of siliceous formations. Volcanic/hydrothermal sources have been identified as the primary sources of dSi for chert formation, not only by us (initially in Jurkowska and Świerczewska-Gładysz, 2020a, b - not Jurkowska, 2022), but by many other researchers. For details, please see our review: Jurkowska and Świerczewska-Gładysz, 2024. We also include a summary of all presented models of chert formation and dSi sources in Jurkowska and Świerczewska-Gładysz, 2020. To clarify the situation, we have added the descriptions and the cited literature cited therein. We wish to note that our model has been published in high-ranking journals, was thoroughly reviewed by several experts, and is supported by the significant scientific presented in our articles.*

Reviewer: In today's oceans only 11% of the total dSi inputs comes from volcano-hydrothermal sources (Treguer et al. 2021).

*Authors: We agree, but this situation is valid only for recent times and does not have any impact on the Cretaceous. During the Paleozoic and Mesozoic, the magnitude of dSi released from volcano-hydrothermal sources was several times higher than today due to plate tectonic processes (e.g. compare the rate of spreading in the Cretaceous with today). We find some difficulty in responding to these suggestions because they have nothing in common with the Si circulation in the time interval that we are studying. We never assume that our model of Si cycle is valid for current environment. In our previous works, we also added a chapter comparing the Cretaceous Si circulation with the recent Si cycle (Tréguer et al., 2021).*

Reviewer: The inputs are only a small amount of dSi relative to the total burden of dSi in the water of the entire oceans. I fail to see how this could be a major factor.

*Authors: What does "Total burden of dSi in a water" mean?*

Reviewer: The other complicating factor is the diffusion of these supposedly high dSi concentrations from the water column into the sediment. How far into the sediment do you envision the dSi penetrating? Usually dSi is high in sediments and the diffusion will be working against a concentration gradient.

*Authors: Considering the distinctive nature of the Cretaceous seabed mud, which had no modern equivalents, and consisted of highly porous and permeable, non-lithified coccoliths mud with high porewater content, dSi diffusion could extend up to the sulphate reduction zone. All processes of dSi diffusion and silica polymorphs precipitation occurred during early diagenesis, before sediment lithification, with no evidence of late diagenesis or secondary infiltration of dSi-reach porewater during late diagenesis. We discuss the detailed model of dSi diffusion based on the Landmesser diffusion model in Jurkowska and Świerczewska-Gładysz, 2020, and here we present a dynamic model of dSi diffusion within the sediment among three working sites: sweater/porewater, decaying sponge skeletons, and newly forming chert nodules.*

**To address the above comments regarding the modern Si cycle and circulation and to avoid any misleading interpretation, we will add a chapter comparing the modern Si cycle (Treguer et al., 2021) with the pre-Eocene Si cycle. The main reason for adding this chapter is to highlight the differences between dSi sources, burial, and circulation in these two distinctly different contexts.**

Reviewer: Overall, the structure is not well organized. For example, Section 2 and Section 3 are essentially literature reviews without new work, so it should be included into the Introduction. Section 5.1 is a description of results, while Section 5.2-5.3 is mostly discussion. It will be more readable to describe the observations of sponge spicules, organic matter and minerals in this Section, and move subjective investigations to the following Section Discussion. The authors even wrongly labeled section 5.3.3 (Line 595), which I assume should be 5.4? The manuscript is currently too confusing and wordy in the present version.

*Authors: Section 2 is descriptive and based on the literature. The main reason for putting these chapter is the fact that SE is a journal which is addressed to the broad audience of researchers and we are aware that not all of them are familiar with geological sciences. If we add this chapter into introduction this will be too long and the main idea of the introduction section which is presenting the main goals will be lost. Section 3 is an description of methodological approach, explaining what was the main idea of samples and sections selection and this is definitely a new work (it is not easy to find the sections which has been described in terms of stratigraphy, lithology and the collections of siliceous sponges are available). Taking into account the suggestion we decide to include that section in chapter about the materials and methods. The number of a chapter 5.3.3 will be corrected. Regarding the reorganization of the chapters, we will follow the suggestion made by the Reviewer. The results section, discusses the mineralogy of sponge spicules (Chapter 5.1), will be supplemented with a description of OM content. Meanwhile, Chapters 5.2-5.3 will be incorporated into the discussion section.*

Reviewer: The manuscript includes repeated discussion and sometimes self-contradictory statements. For example, in lines 427-428, they state "taking into account that during the Cretaceous the seawater dSi concentration was high (Siever, 1991)", in lines 439-443 they also

Reviewer: claim "similar to those... under relatively high dSi... contradicts the diminished seawater dSi", but then in lines 707-708 they state "low seawater dSi concentration is very probable". I got lost several times when reading the text, a few concluding sentences in each section may help to understand their main idea.

*Authors: Those sentences will be clarified and detailed (in numbers ppm/μM)to avoid any doubts.*

Reviewer: I agree that the dSi concentration of porewater is important for silica precipitation, but how to exclude the dissolution of spicules as part of the porewater after burial? The dissolution pits on the spicule surface and the voids left by spicules indicate that dissolution is happening, and dSi can diffuse to the surrounding matrix.

*Authors: Yes, we agree that the dissolution of spicules took place in studied sediments and the dSi from spicules saturated porewaters. Unfortunately, the geochemical analysis that could distinguish and calculate the percentage of that inflow does not exist (although we are still trying various geochemical analyses). That is why we used mineralogical and microtextural studies to answer this question. In the presented ms, we performed our studies in a section at Folwark quarry, where siliceous sponges occur, but no silica polymorphs were found. Our studies also indicated that the facies pattern of correlative occurrence of siliceous sponges and cherts is not supported by geological record(Jurkowska and Świerczewska-Gładysz, 2020a,b; 2024), which means that cherts were formed in a regions where silicifiers were rare. The microtextural studies revealed that the chert boundaries do not follow the sponge outlines (Chapter 5.2). The preservation of sponge skeletons as siliceous and as voids left after spicules, but always covered by layer of opal-CT in a rock that contains opal-CT in the matrix, indicates a constant dSi inflow, which took place over a long time (considering the thickness of the studied successions estimated to be about 500 m).*

Reviewer: The reprecipitation of silica (opal-CT or quartz) in some voids may consume comparable or a bit higher amount of original silica, but what about the voids occupied by pyrite, barite or not occupied at all? Where is the original spicule Si? Isn't this a dSi source for porewater?

*Authors: The voids occupied by pyrite/marcasite, which we describe in this ms from the Turonian of the Folwark quarry. This section has never been studied in detail for their mineralogy and elemental composition or the seawater dSi concentration. Considering that in the Cretaceous European Basin the dSi concentrations were not consistently high (Jurkwoska et al., 2019), two scenarios are possible (which we describe in Chapter: 5.3) under the low dSi concentration (below quartz), the dSi from sponge dissolution could saturate the porewater but to a level below the opal-CT/ quartz precipitation levels (since no silica polymorphs have been detected in the rock matrix), or as in same specimens studied, due to high clays content, the clays were scavening dSi during their diagenetic transformations, or the clays protected the dSi outflow from the sponge and from the secondary infilling of the spicules (Fig. 8 in this ms). The complexity of this answer indicates that the dSi concentration is a factor which should be*

*recognized from the analysis of the whole system of dSi diffusion between different sites, which can act as sources or sinks.*

Reviewer: Finally, how should one distinguish if the dSi origins are from seawater or hydrothermal or spicules? How can you estimate the share of each dSi source? How is it reflected in silicon isotopes?

*Authors: Those questions are now the subject of our studies, which involves the $\delta^{30}Si$, analysis, elemental composition (Al, Ti, Mg, REE+Y) and opal-CT growing lab experiments. We are not able to answer on them yet. The method that has been widely used in geological studies for distinguishing hydrothermal and seawater dSi sources (in Precambrian cherts, before silicifiers become abundant) is summarized in Jurkowska and Świerczewska-Gładysz, 2024 and the literature cited therein (tab. 1) and includes: $\delta^{30}Si$, REE +Y, Al-Fe-Mn ternary diagram, Hg enrichments, Eu/Eu$^*$ anomaly.*

Specific comments

Reviewer: Line 1. The title. Can you please define what time period is "pre-Ecocene"? What does "perspective rock mineralogy" mean? Why is "perspective" in the title?

*Authors: The phrase "pre-Eocene Si cycle" refers to the model of the Si cycle that occurred during Paleozoic and Mesozoic. We are planning to add a chapter about the evolution of the Si cycle in Earth's history at the begging, so this term will be clarified. The perspective is to highlight the analytical method used, namely mineralogy and microtexture, not geochemistry.*

Reviewer: Lines 13-15. Why use "Both ideas"? Isn't "chert formation" "part of the marine Si cycle"?

*Authors: The chertification process is a part of Si burial and thus integral to the Si cycle. However, our studies focus on a specific seminal model of chert formation (Maliva and Siever, 1989a), which does not associate chert formation with the Si cycle.*

Reviewer: Lines 78-114, 333-373, 400-474, 517-577, 638-671. These paragraphs are excessively large sometimes covering 3 pages of text and should be rewritten.

*Authors: Following the suggestion to reorganize the ms, this section will be moved to the results and rewritten.*

Reviewer: Line 150. What is the definition of "Earth history"?

*Authors: This term is common in geological language and includes all biological and abiotic events that have been described based on the geological record and dated using stratigraphy.*

Reviewer: Line 210. A map or lithological column, as well as some outcrop photos can be added in this section to help readers understand where the samples are collected from and what the samples look like. Moreover, I suggest adding some background information about other silicifiers such as diatoms or radiolarians in the study area.

*Authors: All the sections have been described in detail in our previous articles, as have the rocks in terms of microfacies and mineralogical composition. Including the same description here would only repeat information and extend the text; instead, we cite the literature sources where the reader can find all the geological information and detailed petrographic descriptions of the rocks. The situation is similar for the silicifiers, which were discussed in previous work; however, we will also include this information here.*

Reviewer: Lines 343-348. I can not follow the logic here. Why would the chert nodules overlap with sponges if sponge spicules serve as a dSi source? The different morphology of opal-CT in spicules and the matrix may refer to differences in the microenvironment, and may indicate their independence in the precipitation process, but not the dissolution.

*Authors: If the sponge is the source for chert formation, considering Ostwald rules of silica precipitation, the newly formed lepispheres should follow the outline of the source (for a detailed mechanism, see Jurkowska and Świerczewska-Gładysz, 2020b and the literature cited therein). We will rewrite this sentence and add explanation. In the carbonate Cretaceous mud of carbonate sections with siliceous nodules, the microenvironment was uniform in the sediment and, even around the decaying sponges (with rigid skeletons) formed semi-closed microenvironments. Among the sponges, the non-lithistid demosponges delivered much more dSi compared to other sponges. They spicules were incorporated into the sediment mud after the sponge's death, and considering the distinctive character of the Cretaceous seabed mud in which the geochemical conditions were unified, making it improbable that cherts and spicules were under different geochemical condition in a variable microenvironment. If such a situation would occur, it would be reflected in the mineralogy and microtextures of minerals as described in our studies on Turonian rocks (where no cherts were found). We specialize in early diagenesis processes that take place in the seabed mud. By studying the mineralogy and miecrotexture of the minerals, we are able to reconstruct all the processes that occurred in the sediment.*

Reviewer: Line 350. Does the "seawater" here mean the overlying seawater? To which depth can it react with porewater? How to recognize in which stage of diagenesis, the dissolution or precipitation of spicules happen in the sediments?

*Authors: Seawater generally refers to the water that overlies the sediment (Jurkowska, 2022). All the processes of silica polymorph transformations (dissolution and precipitation) and other authigenic mineral precipitations (calcite, pyrite, barite, marcasite) occur when the necessary factors are available under the specific geochemical conditions. We are able to reconstruct all these processes by studying the mineralogy (quantitative and qualitative), microtexture, and cement stratigraphy. In terms of silica, all the necessary factors: dSi concentration, Mg ions, and alkalinity (please refer to Chapter 5.2 of the ms) can be achieved under certain geochemical zone which are established in the sediment column due to the universal process of microbial organic matter decomposition (Chapter 5.2). The organic matter decomposition is responsible for early diagenetic mineral transformations. We described this process in detail in Jurkowska (2022; also the literature cited therein. We also include a short description of this processes in the ms. i are not able to describe here all the models and complex processes we*

*are based on, but we provide proper citations of the articles in which we also summarize other researchers' ideas and models related to the topic.*

Reviewer: Lines 464-465. The absence of OM does not necessarily mean no decomposition, instead it might indicate complete decay of OM in the oxic zone.

*Authors: Yes, but in the lines mentioned above, we are not discussing the OM in the sediment, but the OM that was preserved in the sponge ("In limestone, the presence of OM associated only with siliceous sponges (Fig. 5f) and with 463 pyrite mineralogy indicates that due to oligotrophic conditions, the OM underwent anaerobic 464 microbial decomposition, while not decaying in an oxic zone").*

Reviewer: Lines 603-604. What kind of transformation?

*Authors: We will add an explanation. The transformation of the cay during diagenesis includes the formation of authigenic clay or the transformation of pre-existing detrital clays (MacKenzie and Garrels, 1967; Siever and Woodford, 1973; Isson and Planavsky, 2018).*

Reviewer: Line 614. Which silica polymorph?

*Authors: We will add an explanation: opal-CT.*

---

## Author Comment (AC4)

Reviewer: In their manuscript, Jurkowska and colleagues examine the factors governing the siliceous sponge dissolution and reprecipitation of silica polymorphs in the seabed mud. For this purpose, they analysed the mineralogy and micro-texture of siliceous (cherts) and carbonate-siliceous (opoka) rocks of the Late Cretaceous and compared those to rocks in which siliceous sponges were found but no silica polymorphs. They concluded that the formation of cherts and silica polymorphs is an open system with a dynamic diffusion between seawater, seabed mud/ pore water and sponge.

Authors: Exactly. Thank you for the very insightful summary.

Reviewer: The study also raises the question what information is recorded in the Si isotope data of these siliceous rocks, which is important for future interpretations of these signals.

Authors: We agree and wish to emphasize that, with regard to the $\delta^{30}$Si isotope study, we are merely raising a question rather than questioning the method, as has been suggested by the Referee #1.

Reviewer: Generally, the structure of the MS should be improved, as pointed out by anonymous referee #1. As proposed and planned by the authors, the "re-distribution" of section 5 and the incorporation of section 3 into section 4 "materials and methods" will be helpful. Anonymous referee #1 pointed out the manuscript tends to be too "confusing and wordy". I agree that the authors can be more concise and provided examples out in the specific comment section.

Authors: The ms will be reorganized according to the structure suggested by both Referees. Additionally, the terminology will be refined and more clearly defined.

Reviewer: Overall, there are some other points to raise, which should be considered more carefully in the revised version. In my opinion the authors statement of "dSi as primary environmental factor" is an oversimplification. The auhtors don't explain what governs the dSi concentration, factors such as pH, temperature, ionic strength, dissolved Si polymers and complexation, seawater and more importantly porewater composition are mentioned to some extent in the manuscript, but I recommend to clearly state how these are related to the Si concentration and how they drive dissolution and precipitation of opal-A..

Authors: The description of the factors driving opal-A dissolution and precipitation will be added to the text. We discuss the factors that drive dSi precipitation from porewater because these are the only parameters we are able to recognize based on geological/mineralogical studies of rocks. The subject of this study is the Cretaceous environment, and the estimations of dSi concentration are based on the study of silica polymorphs preserved in a rock. We are not able to address the factors governing dSi concentration in seawater as outlined by Referee #2 because we lack the knowledge

(from the literature) of these factors and we cannot reconstruct them based on rock studies alone. The only material we have for study is the rock, which was formed through the sedimentation of material on the seabed followed by diagenesis. Based on previous studies (Siever, 1991) and our models, we estimate the dSi concentration in seawater, which was likely variable within the water column and exhibited seasonal changes. However, we cannot definitively determinate this from geological studies alone. Unfortunately, geological studies of rocks that are 70 million years old face significant limitations, and we are unable to apply the same analyses used for modern or younger sediment studies. To clarify these differences we will add a chapter describing the limitations inherent in geological studies of Cretaceous and older rocks.

Reviewer: Lastly, I would recommend to also cite more studies from the recent years, as there are a few investigating the dissolution and precipitation behaviour of opal-A under various conditions. Even though these are often experimental studies, I believe this would strengthen the authors arguments. Further points to consider are, that not only Mg, but also Al and Fe influence the solubility of opal-A.

Authors: We will add a discussion on the variable patterns of opal-A dissolution and the factors controlling it.

Specific comments

Reviewer: L37-42: These sentences are almost identical. Here I recommend being more concise.

Authors: The sentence will be changed.

Reviewer: L78-93: The structure of the paragraph could be improved by e.g. starting with the description of the classical model and then point out that it is derived by mineralogical and paleontological analysis. Within this paragraph, L87: The authors state that the classical model is generally accepted and extended for other siliceous rocks like opoka. The literature given regarding the "other siliceous rocks (e.g. opoka...)" is dated significantly earlier than that of Maliva and Siever, 1989a,b., which would imply these studies developed the model. I am aware this is a detail, and just a matter of wording, but could lead to confusion.

Authors: Thank you for that comment. We will improve this chapter in line with the suggestions.

Reviewer: L97: Some of the "many studies" should be cited.

Authors: The reference will be added to the text.

Reviewer: L120: What do the authors mean with "geochemically dependent"?

Reviewer: L126-130: Impurities (e.g., Al) within the opal structure of the sponge could also affect its solubility.

Authors: This means that it is controlled by geochemical conditions. Following the general comment of the Referee, this section will be extended and explained in detail, including recent studies on the factors controlling opal-A dissolution.

Reviewer: L131-136: This sentence is long and hard to follow.

Authors: The sentence will be changed.

Reviewer: L166-171: These statements need a reference.

Authors: The sentence will be rewritten.

Reviewer: Another point to consider in this paragraph or later on in the MS (section 5.3 maybe) is the nature of OM. Amino acids significantly enhance the dissolution of amorphous silica (https://doi.org/10.1346/CCMN.2009.0570203), which could also affect preservation of the siliceous material.

Authors: In this chapter, we discuss our methodological approach, which includes the general environmental trophic conditions in terms of the presence of OM undergoing microbial decomposition, which generates geochemical zonation within the seabed mud, triggering silica precipitation. We did not analyze the nature of the OM, but in the article by Kawano et al. (2009), the authors postulate that in specific solutions containing various amino acids at pH 4, these acids can enhance the dissolution of opal-A. This situation could occurred in the close environment of sponges in the Turonian marls we studied, and we will incorporate this suggestion and discuss other factors affecting the rate of opal-A dissolution. The scale of this process is difficult to estimate because, in the studied fossils preserved as pyrite/marcasite, the pH would not drop below 6, as the primary calcium carbonate is not dissolved.

Reviewer: L190-193: The dynamics of this process should be presented more precise and by emphasising that clay formation scavenges dSi out of the water and precipitate in the voids. dSi concentration is therefore low, which would slow down the reprecipitation process etc.

Authors: We will add a more detailed description of the process of dSi scavenging by clays from the porewaters.

Reviewer: L278: Section 5 contains many parts where the sentences are very long and complicated which makes it often hard to follow. This should be considered when incorporating this section into the discussion.

Authors: Section 5 will be incorporated into the discussion, and the lengthy sentences will be rewritten.

Reviewer: L348-535: This sentence is long and I have trouble following the authors argument here, and I am not sure how it is connected to the previous sentence.

Authors: The previous sentence has been rewritten as suggested by Referee #1. The long sentence has also been revised.

Reviewer: L394: If "Mg-rich clays" form or a precursor, there must be Al present which affects the solubility of amorphous Si significantly, which has so far not been discussed.

Authors: The Mg-rich clays are a very early phase that precipitates simultaneously with silica polymorphs (opal-CT) and is likely a precursor phase of opal-CT (Kastner et al., 1977). The presence of this early form does not indicate significantly high Al content, but is presence in an environment in casual amount (Jurkowska et al., 2019). During precipitation, early silica forms incorporate foreign ions into their structure, including naturally occurring ions in porewater such as Al and Mg.

Reviewer: L608-610: Please add the reference of this experimental study.

Authors: The reference will be added to the text.

Reviewer: L702-712: In various studies it has been shown that in the sediment the porewater dSi increases with depth. To what sediment depth do you assume the seawater influences the mud. What depth do you assume the diagenesis took place?

Authors: In modern sediments, dSi increases with depth due to the dissolution opal-A. As we highlighted in our response to Referee #1, the Cretaceous seabed mud was very distinctive in composition and characteristics. Although we are unable to estimate the exact depths within the sediment, we can reconstruct the geochemical zonation based on the presence of authigenic minerals and geochemical conditions required for their precipitation). The Cretaceous seabed mud was non-lithified, water-saturated, with a pudding-like consistency, and exhibited high porosity and permeability. Careful estimations suggest that the SRZ zone was located at a depth of 25cm-2m depth (Clayton, 1984).

Reviewer: Temperature would influence Si solubility levels and maybe responsible for the lack of silica polymorph precipitation. What is the role of temperature in their system? What depth did the diagenesis occur in, did the elevated seawater temperature of the Cretaceous influence the system?

Authors: The temperature could affect the transformation of silica polymorphs during late diagenesis (due to an increase temperature with burial), but this did not occur in

the studied sediment, which were only shallowly buried and did not reach the temperatures high enough to trigger the diagenetic changes in the silica polymorphs (Jurkowska and Świerczewska-Gładysz, 2020a). Had such changes occurred, we would have observed them not only through mineralogical and microtextural alternations of the silica polymorphs but also in other minerals. Another effect of temperature on silica polymorphs could be inducted by seawater temperature. Unfortunately, we have limited data on the temperature of the seawater near the seabed. Bojanowski et al. (2016) studied the Campanian chalk of Poland and estimated that the near-surface temperatures could range from approximately 16-19°C, while bottom and porewater were around 13-18°C. However, these data reflect only approximate temperatures for a specific time interval and cannot be interpreted as permanent. The studied area was influenced by both colder Atlantic waters and warmer Tethyan currents, but the detailed circulation pattern during the studied interval has only been briefly recognized (Remin et al., 2016). According to our model, the main source of dSi originates from the LIP sources in the Atlantic ocean and was delivered to the European Basin by inflows, suggesting that the siliceous rocks may have formed under the colder water inflows. However, to trace this further, we would need to perform $\delta^{18}O$ analysis of the silica polymorphs. The potential influence of temperature on silica polymorph precipitation was discussed in our previous studies (Jurkowska and Świerczewska-Gładysz, 2020a; b).

Technical corrections

Reviewer: L80: Cenozoic

Authors: The sentence has been changed and do not include that word.

Reviewer: L206: Remove one comma after in contrast.

Authors: It will be corrected.

Reviewer: L236: has been investigated in previous studies

Authors: It will be corrected.

Reviewer: L242: voids of them

Authors: It will be corrected.

Reviewer: L357-359: I strongly recommend simplifying this part of the sentence for the reader

Authors: It will be corrected.

Reviewer: L358: "remarks" seems wrong here

Reviewer: L395: Do you mean dissolution marks?

Authors: Should be "marks". It will be corrected.